# LiveBench: A Challenging, Contamination-Limited LLM Benchmark

**Colin White**[*1]**, Samuel Dooley**[*1]**, Manley Roberts**[*1]**, Arka Pal**[*1]**, Benjamin Feuer**[2]**,
**Siddhartha Jain**[3]**, Ravid Shwartz-Ziv**[2]**, Neel Jain**[4]**, Khalid Saifullah**[4]**, Sreemanti Dey**[1]**,
**Shubh-Agrawal**[1]**, Sandeep Singh Sandha**[1]**, Siddartha Naidu**[1]**, Chinmay Hegde**[2]**,
**Yann LeCun**[2]**, Tom Goldstein**[4]**, Willie Neiswanger**[5]**, Micah Goldblum**[6]

[1] Abacus.AI, [2] NYU, [3] Nvidia, [4] UMD, [5] USC, [6] Columbia

https://livebench.ai

## ABSTRACT

Test set contamination, wherein test data from a benchmark ends up in a newer model's training set, is a well-documented obstacle for fair LLM evaluation and can quickly render benchmarks obsolete. To mitigate this, many recent benchmarks crowdsource new prompts and evaluations from human or LLM judges; however, these can introduce significant biases, and break down when scoring hard questions. In this work, we introduce a new benchmark for LLMs designed to be resistant to both test set contamination and the pitfalls of LLM judging and human crowdsourcing. We release LiveBench, the first benchmark that (1) contains frequently-updated questions from recent information sources, (2) scores answers automatically according to objective ground-truth values, and (3) contains a wide variety of challenging tasks, spanning math, coding, reasoning, language, instruction following, and data analysis. To achieve this, LiveBench contains questions that are based on recently-released math competitions, arXiv papers, news articles, and datasets, and it contains harder, contamination-limited versions of tasks from previous benchmarks such as Big-Bench Hard, AMPS, and IFEval. We evaluate many prominent closed-source models, as well as dozens of open-source models ranging from 0.5B to 405B in size. LiveBench is difficult, with top models achieving below 70% accuracy. We release all questions, code, and model answers. Questions are added and updated on a monthly basis, and we release new tasks and harder versions of tasks over time so that LiveBench can distinguish between the capabilities of LLMs as they improve in the future. We welcome community engagement and collaboration for expanding the benchmark tasks and models.

## 1 INTRODUCTION

In recent years, as large language models (LLMs) have risen in prominence, it has become increasingly clear that traditional machine learning benchmark frameworks are no longer sufficient to evaluate new models. Benchmarks are typically published on the internet, and most modern LLMs include large swaths of the internet in their training data. If the LLM has seen the questions of a benchmark during training, its performance on that benchmark will be artificially inflated (referred to as "test set contamination") (Roberts et al., 2024; Dong et al., 2024; Deng et al., 2023; Golchin & Surdeanu, 2023b), hence making many LLM benchmarks unreliable. Recent evidence of test set contamination includes the observation that LLMs' performance on Codeforces plummet after the training cutoff date of the LLM (Roberts et al., 2024; Jain et al., 2024), and before the cutoff date, performance is highly correlated with the number of times the problem appears on GitHub (Roberts et al., 2024). Similarly, a recent hand-crafted variant of the established math dataset, GSM8K, shows evidence that several models have overfit to this benchmark (Zhang et al., 2024; Cobbe et al., 2021).

To lessen dataset contamination, benchmarks using LLM or human prompting and judging have become increasingly popular (Jain et al., 2024; Chiang et al., 2024; Zheng et al., 2024; Li et al., 2024). However, using these techniques comes with significant downsides. While LLM judges have

---

*crwhite@meta.com, dooley@meta.com, mig2132@columbia.edu. Sponsored by Abacus.AI.

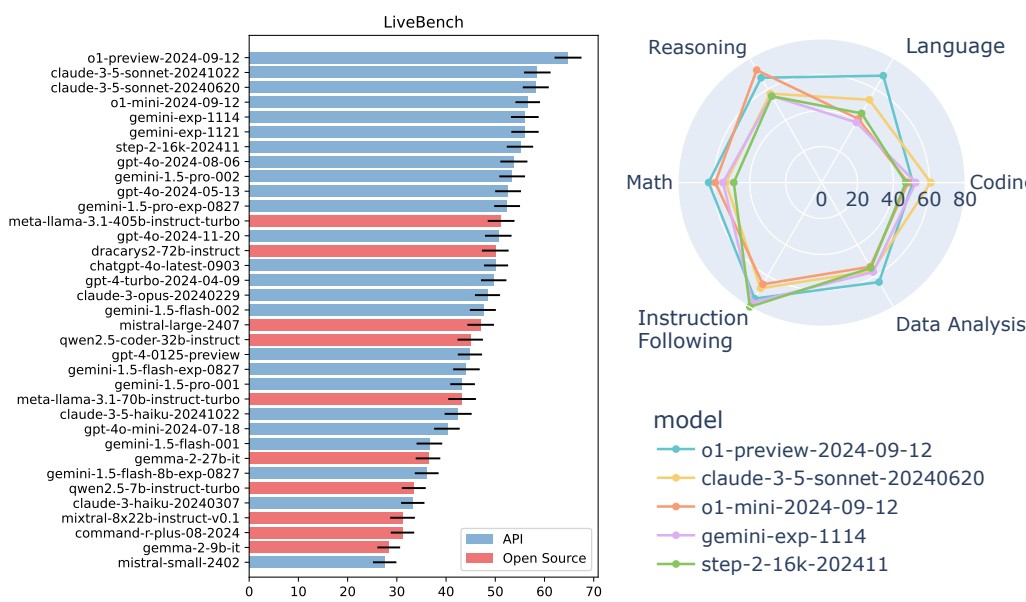

Figure 1: Left: results on `LiveBench` for top models, showing 95% bootstrap confidence intervals. Right: a radar plot for select models across `LiveBench`'s six categories demonstrating the that ordering of top models varies between each category.

multiple advantages, such as their speed and ability to evaluate open-ended questions, they are prone to making mistakes and can have several biases (Li et al., 2024). Furthermore, LLMs often favor their own answers over other LLMs, and LLMs favor more verbose answers (Li et al., 2024; Dubois et al., 2024; Li et al., 2023b). Additionally, using humans to provide evaluations of LLMs can inject biases such as formatting of the output and the tone of the writing (Chiang et al., 2024). Using humans to generate questions also presents limitations. Human participants might not ask diverse questions, may favor certain topics that do not probe a model's general capabilities, or may construct their prompts poorly (Zheng et al., 2024).

In this work, we introduce a framework for benchmarking LLMs designed to minimize both test set contamination and the pitfalls of LLM judging and human crowdsourcing. We use this framework to create `LiveBench`, the first benchmark with these three desiderata: *(1)* `LiveBench` contains frequently-updated questions based on recent information sources; *(2)* `LiveBench` is scored automatically according to the objective ground truth without the use of an LLM judge; and *(3)* `LiveBench` questions are drawn from a diverse set of six categories. We ensure *(2)* by only including questions that have an objectively correct answer. `LiveBench` questions are *difficult*: no current model achieves higher than 70% accuracy. Questions are added and updated on a monthly basis, and we release new tasks and harder versions of tasks over time so that `LiveBench` can distinguish among the capabilities of LLMs as they improve in the future.

**Overview of tasks.**  `LiveBench` currently consists of 18 tasks across 6 categories: math, coding, reasoning, language, instruction following, and data analysis. Each task falls into one of two types: *(1)* tasks which use an information source for their questions, e.g., data analysis questions based on recent Kaggle datasets, or fixing typos in recent arXiv abstracts; and *(2)* tasks which are more challenging or diverse versions of existing benchmark tasks, e.g., from Big-Bench Hard (Suzgun et al., 2023) or IFEval (Zhou et al., 2023a). The categories and tasks included in `LiveBench` are:

- **Math:** modified questions based on high school math competitions from the past 11 months, as well as harder versions of AMPS questions (Hendrycks et al., 2021)
- **Coding:** code generation questions from recent Leetcode and AtCoder questions (via Live-CodeBench (Jain et al., 2024)), as well as a novel code completion task
- **Reasoning:** a harder version of Web of Lies from Big-Bench Hard (Suzgun et al., 2023), and novel Zebra Puzzles (e.g., (Jeremy, 2009)) and spatial reasoning questions

- **Language Comprehension:** Connections word puzzles, a typo-fixing task, and a movie synopsis unscrambling task for recent movies on IMDb and Wikipedia
- **Instruction Following:** four tasks to paraphrase, simplify, summarize, or generate stories about recent new articles from The Guardian (Guardian Media Group, 1821), subject to one or more instructions such as word limits or incorporating specific elements in the response
- **Data Analysis:** three tasks using recent datasets from Kaggle and Socrata, specifically, table reformatting (among JSON, JSONL, Markdown, CSV, TSV, and HTML), predicting which columns can be used to join two tables, and predicting the correct column type annotation

We evaluate dozens of models, including proprietary models as well as open-source models with sizes ranging from 0.5B to 8x22B. We release all questions, code, and model answers, and we welcome community engagement and collaboration. Our codebase is available at `https://github.com/livebench/livebench`, and our leaderboard is available at `https://livebench.ai`.

## 2 LIVEBENCH DESCRIPTION

In this section, we introduce `LiveBench`. It currently has six categories: math, coding, reasoning, data analysis, instruction following, and language comprehension. Categories are diverse with two to four tasks per problem. Each task either includes recent information sources (such as very recent news articles, movie synopses, or datasets) or is a more challenging, more diverse version of an existing benchmark task.

Each task is designed to have 40-100 questions which span a range of difficulty, from easy to very challenging, while loosely aiming for an overall 30-70% success rate on the top models for each task. Prompts are tailored for each category and task but typically include the following: zero-shot chain of thought (Kojima et al., 2022; Wei et al., 2022), asking the model to make its best guess if it does not know the answer, and asking the LLM to output its final answer in a way that is easy to parse, such as in XML tags or in \*\*double asterisks\*\*. We also acknowledge that parsing answers in this way requires some degree of instruction following, and we address this in Appendix A.4. In the following sections, we give a summary of each task from each category. See Appendix A.3 for additional details.

### 2.1 MATH CATEGORY

Evaluating the mathematical abilities of LLMs has been one of the cornerstones of recent research in LLMs, featuring prominently in many releases and reports (Reid et al., 2024; OpenAI, 2023; Bubeck et al., 2023). Our benchmark includes math questions of three types: questions modified from recent high school math competitions, fill-in-the-blank questions from recent olympiad competitions, and questions from our new, harder version of the AMPS dataset (Hendrycks et al., 2021).

Our first two math tasks, `Competitions` and `Olympiad`, are based on expert human-designed math problems that offer a wide variety in terms of problem type and solution technique. In `Competitions`, we include questions from AMC12 2023, SMC 2023, and AIME 2024 modifying the prose and the answer order; in `Olympiad`, we include questions based on USAMO 2024 and IMO 2024, in which the task is to rearrange masked out equations from the solution into the correct order. These questions test problem solving with algebra, combinatorics, geometry, logic, number theory, probability, and other secondary school math topics (Faires & Wells, 2022).

Finally, we release synthetically generated math questions in the `AMPS_Hard` task. This task is inspired by the math question generation used to create the MATH and AMPS datasets (Hendrycks et al., 2021). We generate harder questions by drawing random primitives, using a larger and more challenging distribution than AMPS across 10 of the hardest tasks within AMPS.

### 2.2 CODING CATEGORY

The coding ability of LLMs is one of the most widely studied and sought-after skills for LLMs (Mnih et al., 2015; Jain et al., 2024; Li et al., 2023a). We include two coding tasks in `LiveBench`: a modified version of the code generation task from LiveCodeBench (LCB) (Jain et al., 2024), and a novel code completion task combining LCB problems with partial solutions collected from GitHub.

The `LCB Generation` assesses a model's ability to parse a competition coding question statement and write a correct answer. We include 78 questions from LiveCodeBench (Jain et al., 2024) which has several tasks to assess the coding capabilities of large language models.

The `Completion` task specifically focuses on the ability of models to complete a partially correct solution—assessing whether a model can parse the question, identify the function of the existing code, and determine how to complete it. We use LeetCode easy, medium, and hard problems from LiveCodeBench's (Jain et al., 2024) April 2024 release, combined with matching solutions from https://github.com/kamyu104/LeetCode-Solutions, omitting the last 15-70% of each solution and asking the LLM to complete the solution.

## 2.3 REASONING CATEGORY

The reasoning ability of large language models is another highly benchmarked and analyzed skill of LLMs (Wei et al., 2022; Suzgun et al., 2023; Yao et al., 2024). In `LiveBench`, we include three reasoning tasks: our harder version of a task from Big-Bench Hard (Suzgun et al., 2023), Zebra puzzles, and spatial reasoning questions.

`Web of Lies v2` is an advancement of the similarly named task included in Big-Bench (bench authors, 2023) and Big-Bench Hard (Suzgun et al., 2023). The task is to evaluate the truth value of a random Boolean function expressed as a natural-language word problem. We create new, significantly harder questions by including additional deductive components and several types of red herrings. Next, we include spatial reasoning questions. This set of 50 handwritten questions tests a model's ability to make deductions about intersections and orientations of common 2D and 3D shapes.

Finally, we include `Zebra Puzzles`, a well-known reasoning task (Jeremy, 2009) that tests the ability of the model to follow a set of statements that set up constraints, and then logically deduce the requested information. We build on an existing repository for procedural generation of Zebra puzzles (quint t, 2023). Below, we provide an example question from the `Zebra Puzzles` task.

> **An example question from the `Zebra Puzzle` task.**
>
> There are 3 people standing in a line numbered 1 through 3 in a left to right order.
> Each person has a set of attributes: Food, Nationality, Hobby.
> The attributes have the following possible values:
> - Food: nectarine, garlic, cucumber
> - Nationality: chinese, japanese, thai
> - Hobby: magic-tricks, filmmaking, puzzles
>  and exactly one person in the line has a given value for an attribute.
>  Given the following premises about the line of people:
> - the person that likes garlic is on the far left
> - the person who is thai is somewhere to the right of the person who likes magic-tricks
> - the person who is chinese is somewhere between the person that likes cucumber and the person who likes puzzles
>  Answer the following question: What is the hobby of the person who is thai? Return your answer as a single word, in the following format: **X**, where X is the answer.

## 2.4 DATA ANALYSIS CATEGORY

LiveBench includes three practical tasks in which the LLM assists in data analysis or data science: column type annotation, table join prediction, and table reformatting. Each question makes use of a recent dataset from Kaggle or Socrata.

The first task is to predict the type of a column of a data table. To create a question for the column type annotation task (`CTA`), we randomly sample a table and randomly sample a column from that table. We use the actual name of that column as the ground truth and then retrieve some samples from that column. We provide the name of all the columns from that table and ask the LLM to select the true column name from those options.

Data analysts often also require a table to be reformatted from one type to another, e.g., from some flavor of JSON to CSV or from XML to TSV. We emulate that task in `TableReformat` by providing a table in one format and asking the LLM to reformat it into the target format.

Finally, another common application of LLMs in data analysis is performing table joins (Goldbloom, 2024; Liu et al., 2024b; Sheetrit et al., 2024). In the `TableJoin` task, the LLM is presented with two tables with partially overlapping sets of columns. The LLM is tasked with creating a valid join mapping from the first to the second table.

## 2.5 INSTRUCTION FOLLOWING CATEGORY

An important ability of an LLM is its capability to follow instructions. To this end, we include instruction-following questions in our benchmark, inspired by IFEval (Zhou et al., 2023a), which is an instruction-following evaluation for LLMs containing verifiable instructions such as "write more than 300 words" or "Finish your response with this exact phrase: {end_phrase}." While IFEval used a list of 25 verifiable instructions, we use a subset of 16 that excludes instructions that do not reflect real-world use-cases. See Appendix Table 3. Furthermore, in contrast to IFEval, which presents only the task and instructions with a simple prompt like "write a travel blog about Japan", we provide the models with an article from The Guardian (Guardian Media Group, 1821), asking the models to adhere to multiple randomly-drawn instructions while asking the model to complete one of four tasks related to the article: `Paraphrase`, `Simplify`, `Story Generation`, and `Summarize`. We score tasks purely by their adherence to the instructions.

## 2.6 LANGUAGE COMPREHENSION CATEGORY

Finally, we include multiple language comprehension tasks. These tasks assess the language model's ability to reason about language itself by, (1) completing word puzzles, (2) fixing misspellings while leaving other stylistic changes in place, and (3) reordering scrambled plots of unknown movies.

First, we include the `Connections` category. Connections is a word puzzle popularized by the New York Times (although similar ideas have existed previously). In this task, we present questions of varying levels of difficulty with 8, 12, and 16-word varieties. The objective of the game is to sort the words into sets of four words, such that each set has a 'connection' between them.

Next, we include the `Typos` task. The idea behind this task is inspired by the common use case for LLMs in which a user asks the LLM to identify typos and misspellings in some written text but to leave other aspects of the text unchanged. We create the questions for this task from recent ArXiv abstracts, which we ensure originally have no typos, by programmatically injecting common human typos into the text. Below is an example question from the `Typos` task.

---

**An example question from the `Typos` task.**

Please output this exact text, with no changes at all except for fixing the misspellings. Please leave all other stylistic decisions like commas and US vs British spellings as in the original text.

We inctroduce a Bayesian estimation approach forther passive localization of an accoustic source in shallow water using a single mobile receiver. The proposed probablistic focalization method estimates the timne-varying source location inther presense of measurement-origin uncertainty. In particular, probabilistic data assocation is performed to match tiome-differences-of-arival (TDOA) observations extracted from the acoustic signal to TDOA predicitons provded by the statistical modle. The performance of our approach is evaluated useing rela acoustic data recorded by a single mobile reciever.

---

Finally, we include the `Plot Unscrambling` task, which takes the plot synopses of recently-released movies from IMDb or Wikipedia. We randomly shuffle the synopses sentences and then ask the LLM to simply reorder the sentences into the original plot. We find that this task is very challenging for LLMs, as it measures their abilities to reason through plausible sequences of events.

## 2.7 LIVEBENCH UPDATES AND MAINTENANCE PLAN

Maintaining a contamination-limited benchmark requires that we update the set of questions over time. We have so far released two updates, and we plan to continue to release updates to add new questions and remove outdated questions. In each update, we replace $1/6$ of the questions on average, so that the benchmark is fully refreshed roughly every 6 months. We may speed up the turnover

Table 1: **LiveBench results across the 15 top-performing models.** We display in this table the highest-performing models on `LiveBench`, outputting the results on each main category, as well as each model's overall performance. See Table 2 for the results on all 40 models.

| Model | LiveBench Score | Coding | Data Analysis | Instruction Following | Language | Math | Reasoning |
|---|---|---|---|---|---|---|---|
| o1-preview-2024-09-12 | 64.7 | 50.8 | **64.0** | 74.6 | **68.7** | **62.9** | 67.4 |
| claude-3-5-sonnet-20241022 | 58.5 | **67.1** | 52.8 | 69.3 | 53.8 | 51.3 | 56.7 |
| claude-3-5-sonnet-20240620 | 58.2 | 60.8 | 56.7 | 68.0 | 53.2 | 53.3 | 57.2 |
| o1-mini-2024-09-12 | 56.7 | 48.1 | 54.1 | 65.4 | 40.9 | 59.2 | **72.3** |
| gemini-exp-1114 | 56.0 | 52.4 | 57.5 | 77.1 | 38.7 | 54.9 | 55.7 |
| gemini-exp-1121 | 56.0 | 50.4 | 57.0 | **80.1** | 40.0 | 62.8 | 45.8 |
| step-2-16k-202411 | 55.1 | 46.9 | 54.9 | 79.9 | 44.5 | 48.9 | 55.5 |
| gpt-4o-2024-08-06 | 53.8 | 51.4 | 52.9 | 68.6 | 47.6 | 48.2 | 53.9 |
| gemini-1.5-pro-002 | 53.4 | 48.8 | 52.3 | 70.8 | 43.3 | 57.4 | 47.9 |
| gpt-4o-2024-05-13 | 52.6 | 49.4 | 52.4 | 68.2 | 50.0 | 46.0 | 49.6 |
| gemini-1.5-pro-exp-0827 | 52.4 | 40.9 | 50.8 | 69.3 | 46.1 | 56.1 | 50.9 |
| meta-llama-3.1-405b-instruct-turbo | 51.1 | 43.8 | 53.5 | 72.8 | 43.2 | 40.5 | 52.8 |
| gpt-4o-2024-11-20 | 50.6 | 46.1 | 47.2 | 64.9 | 47.4 | 42.5 | 55.7 |
| dracarys2-72b-instruct | 50.1 | 56.6 | 49.1 | 65.2 | 33.5 | 50.6 | 45.8 |
| chatgpt-4o-latest-0903 | 50.1 | 47.4 | 48.7 | 66.4 | 45.3 | 42.1 | 50.5 |

rate of questions in the future, based on interest in `LiveBench`. Each month, we do not release the new questions until one month later, so that the public leaderboard always has $1/6$ questions that are private. We choose tasks to update based primarily on two factors: *(1)* the oldest tasks, and *(2)* the currently easiest tasks. In this way, the questions in `LiveBench` will stay new and continue to challenge the most capable LLMs. See additional details, as well as a longer discussion on different forms of contamination, in Appendix A.6.

**Method for sustainability**   One downside in a frequently-updating benchmark is that it requires consistent work and computational resources each month. Therefore, we have a plan in place to ensure its continued success. We maintain the best (or most popular) 40-50 models on the leaderboard so as to avoid an ever-growing list of models to evaluate each month. For example, we maintain about two versions of each model family on the leaderboard (to show the improvement from the most recent version) but no more. This ensures that we have a tractable set of at most 50 models to evaluate on 200 questions each month, which is easily within the computational budgets of the authors' institutions. Additionally, we have already had community contributions which further reduces the computational burden of the authors.

The only other recurring work is to update the questions themselves each month. While we are excited and able to add novel tasks each month, many of the tasks are synthetic and therefore very fast and simple to create a new set of questions based on fresh data (e.g., updating the typos task using brand new arXiv papers). Additionally, we have also seen community engagement here as well.

**Completed monthly updates**   In the first monthly update, we added 50 questions in a new spatial reasoning task, 28 additional coding generation questions, and 12 additional coding completion questions. The total size of the benchmark after this update became 1000. In the second monthly update, we fully updated the math olympiad questions, and we partially updated the math AMPS_Hard and math_comp questions, for 132 replaced questions, to maintain 1000 questions.

## 3   EXPERIMENTS

In this section, first we describe our experimental setup and present full results for 40 LLMs on all 18 tasks of `LiveBench`. Next, we give an empirical comparison of `LiveBench` to existing prominent LLM benchmarks, and finally, we present ablation studies.

**Experimental setup.**   Our experiments include 40 LLMs total, with a mix of top proprietary models, large open-source models, and small open-source models. In particular, for proprietary models, we include OpenAI models such as `o1-preview`, `chatgpt-4o`, and `gpt-4o` (Brown et al., 2020; OpenAI, 2023), Anthropic models such as `claude-3-5-sonnet-20240620`,

Google models such as `gemini-1.5-pro-002` (Reid et al., 2024), and Mistral models such as `mistral-large-2407` (Jiang et al., 2023).

For open-source models, we include models such as `Llama-3.1-405b-instruct`, `Llama-3.1-70b-instruct` (Dubey et al., 2024), `deepseek-v2.5`, (Liu et al., 2024a), `qwen2.5-72b-instruct` (Team, 2024b; Yang et al., 2024), `command-r-plus-08-2024` (Cohere, 2024; Cohere For AI, 2024), `gemma-2-27b-it` (Team, 2024a; Team et al., 2024), `mixtral-8x22b-instruct-v0.1` (Jiang et al., 2023), and `phi-3.5-moe-instruct` (Abdin et al., 2024). See Table 4 for a full list of citations.

For all models and tasks, we perform single-turn evaluation with temperature 0, unless otherwise noted in the model card. All models run with their respective templates from our updated version of FastChat (Zheng et al., 2024). We run all open-source models with `bfloat16`. When running new models, we take care to set up its hyperparameters and chat template as in the model's example code, and we also double check the outputs to make sure that the inference, as well as our automated parsing functions, are working correctly and fairly. See more details in Appendix A.4 and Appendix A.5. For each question, a model receives a score from 0 to 1. For each model, we compute the score on each task as the average of all questions, we compute the score on each of the six categories as the average of all their tasks, and we compute the final `LiveBench` score as the average of all six categories. In Appendix B, we give additional documentation including average input/output tokens and cost to run `LiveBench` for each API model.

## 3.1 DISCUSSION OF RESULTS

We compare all 40 models on `LiveBench` according to the experimental setup described above; see Table 1 and Table 2. We find that `o1-preview-2024-09-12` performs the best overall, 6% better than all other models. `o1-preview-2024-09-12` substantially outperforms all other models in the data analysis, language, and math categories. The next-best model is `claude-3-5-sonnet-20240620`, which far outperforms all other models in the coding category (although `o1-mini` outperforms `claude-3.5` in code generation, `claude-3.5` has the edge in code completion). `o1-mini-2024-09-12` is third overall and is significantly better than all other models in the reasoning category.

The best-performing open-source models are `llama-3.1-405b-instruct` and `qwen2.5-72b-instruct`, which virtually tie with each other and outperform `gpt-4-turbo`. The best-performing small open-source model is `phi-3.5-moe-instruct` (see Table 2): with only 6.6B active parameters, it outperforms `gpt-3.5` and is on par with `mixtral-8x22b`.

## 3.2 CORRELATION ANALYSES

Now we present analyses involving correlation among different categories and tasks. First, we compute the Pearson correlation coefficient among all pairs of categories and tasks in `LiveBench` (see Figure 2). We find that unsurprisingly, math, coding, and reasoning all correlate with one another. Interestingly, language correlates fairly well with data analysis, likely due to both categories including tasks that require the LLM to output a large part of the prompt that is modified in a specific way (e.g., by fixing typos or changing the table format). Surprisingly, instruction following correlates relatively weakly with all other categories. Among tasks, we see that math comp correlates the highest with the average `LiveBench` performance, suggesting that this task is the greatest indicator of overall model performance. This is likely due to these being high-quality, diverse mathematical reasoning questions (which we modified to reduce contamination).

Next, in order to see the strengths and weaknesses of each model, we create a scatterplot of each model's overall `LiveBench` performance vs. performance on a single category or task (Figure 3). By plotting a best fit line and computing the residuals for each model, we can compute which models are outliers in specific categories – that is, models that are disproportionately stronger in a particular category relative to the best fit line. We see that the `o1` and `phi` series of models are outliers in terms of reasoning (Figure 3, left), while some of the `Llama`, `gemini`, and `command-r` models are outliers in terms of instruction following. We present additional details in Appendix A.1, including a table of each model's relative best and worst tasks (computed as the highest and lowest residuals).

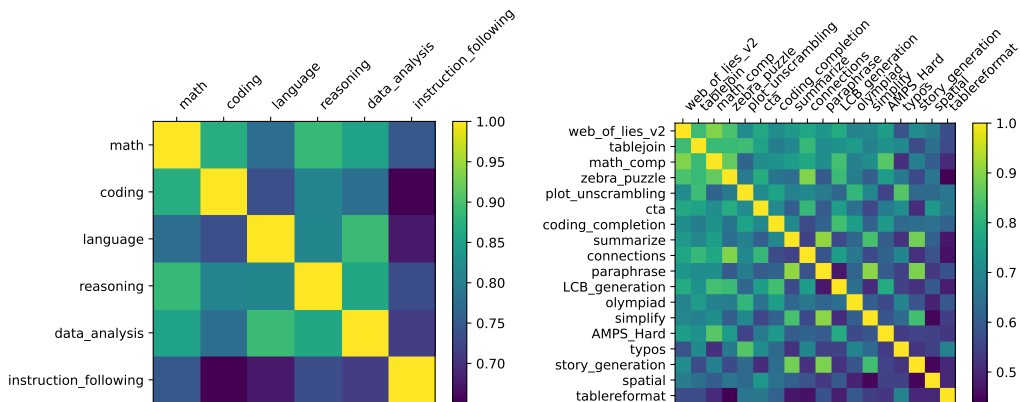

Figure 2: **Correlations among categories and groups in `LiveBench`.** For each pair of categories (left) and tasks (right) in `LiveBench`, we compute the Pearson correlation coefficient based on the results for all 40 models.

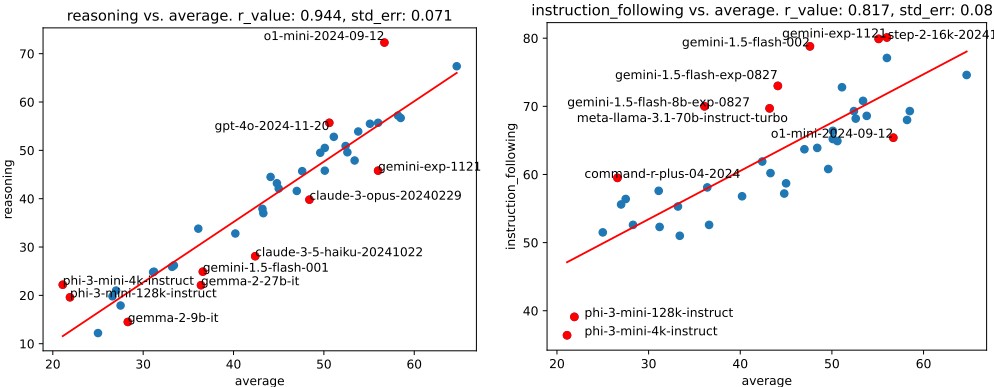

Figure 3: **Reasoning and instruction following performance vs. average performance.** We plot the performance of all 40 models' reasoning (left) and instruction following (right) performance compared to overall `LiveBench` performance, computing a best fit line and annotating outliers.

## 3.3 COMPARISON TO OTHER LLM BENCHMARKS

Next, we compare `LiveBench` to two prominent benchmarks, ChatBot Arena (Chiang et al., 2024) and Arena-Hard (Li et al., 2024). In Figure 4, we show a bar plot comparison among models that are common to both benchmarks, and in Figure 6, we compare the performance of these models to a best-fit line. We also compute the correlation coefficient of model scores among the benchmarks: `LiveBench` has a 0.91 and 0.88 correlation with ChatBot Arena and Arena-Hard, respectively.

Based on the plots and the correlation coefficients, we see that there are generally similar trends to `LiveBench`, yet some models are noticeably stronger on one benchmark vs. the other. For example, `gpt-4-0125-preview` and `gpt-4-turbo-2024-04-09` perform substantially better on Arena-Hard compared to `LiveBench`, likely due to the known bias from using `gpt-4` itself as the LLM judge (Li et al., 2024). We hypothesize that the strong performance of some models such as the `gemini-1.5` models on ChatBot Arena compared to `LiveBench` may be due to having an output style that is preferred by humans. These observations emphasize the benefit of using ground-truth judging, which is immune to biases based on the style of the output.

**Comparison between Ground-Truth and LLM-Judging**    As an additional comparison between `LiveBench` and LLM judge based benchmarks, we give a preliminary study in the Appendix on the efficacy of LLM judging for hard math and reasoning questions. Specifically, we run an initial

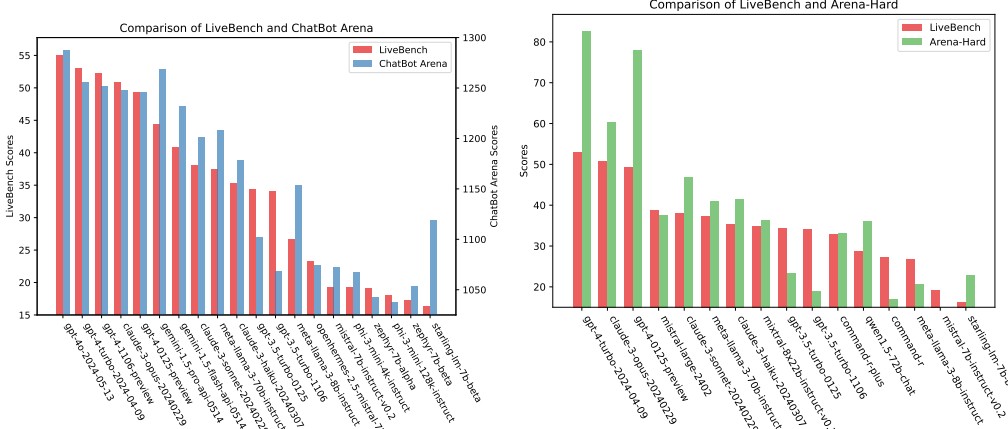

Figure 4: **Comparison of LiveBench to other LLM benchmarks.** We compare `LiveBench` to ChatBot Arena (left) and Arena-Hard (right). We see that while there are generally similar trends, some models are noticeably stronger on one benchmark vs. the other. For example, both GPT-4 models are substantially better on Arena-Hard.

experiment regarding the question, 'if an LLM struggles to answer a hard math or reasoning question, then will the LLM also struggle to determine whether or not a given answer to that question is correct?' Our experiments give evidence that the answer is yes, for zebra puzzles and AMC/AIME questions, but the results are not definitive. See Appendix A.2.

### 3.4 ANALYSIS OF MONTHLY UPDATES

As described in Section 2.7, we have completed two monthly updates for `LiveBench` so far. The rank correlation between the original and first update, and the first and second update, are both $> 0.997$, showing that the model rankings have stayed consistent. On the other hand, between the original and the most-recent set of questions, the median and mean average scores (among models included in all iterations of the leaderboard) have both dropped by about 1.2%, showing that the benchmark is becoming harder over time, as newly released models become more capable.

## 4 RELATED WORK

We describe the most prominent LLM benchmarks and the ones that are most related to our work. For a comprehensive survey, see (Chang et al., 2024). The Huggingface Open LLM Leaderboard (Gao et al., 2021; Beeching et al., 2023) is a widely-used benchmark suite that consists of static datasets such as Big Bench Hard (Suzgun et al., 2023) and MMLU-Pro (Wang et al., 2024). While this has been incredibly useful in tracking the performance of LLMs, its static nature leaves it prone to test set contamination by models.

**LLMs-as-a-judge.** AlpacaEval (Li et al., 2023b; Dubois et al., 2023; 2024), MT-Bench (Chiang et al., 2024), and Arena-Hard (Li et al., 2024) are benchmarks that employ LLM judges on a fixed set of questions. Using an LLM-as-a-judge is fast, relatively cheap, and has the flexibility of being able to evaluate open-ended questions, instruction-following questions, and chatbots. However, LLM judging also has downsides. First, LLMs have biases towards their own answers (Li et al., 2024). In addition, GPT-4 judges have a noticeable difference in terms of variance and favorability of other models compared to Claude judges. Additionally, LLMs make errors. As one example, question 2 in Arena-Hard asks a model to write a C++ program, yet GPT-4 incorrectly judges `gpt-4-0314`'s solution as incorrect (Li et al., 2024).

**Humans-as-a-judge.** ChatBot Arena (Chiang et al., 2024; Zheng et al., 2024) leverages human prompting and feedback. Users ask questions and receive outputs of two randomly selected models and pick which output they prefer. This preference feedback is aggregated into an Elo score for each model. While human evaluation is great for capturing the preferences of a crowd, using a human-

as-a-judge has disadvantages. First, human-judging can be labor-intensive, especially for certain tasks included in `LiveBench` such as complex math, coding, or long-context reasoning problems. Whenever humans are involved in annotation (of which judging is a sub-case), design choices or factors can cause high error rates (Lease, 2011), and even in well-designed human-annotation setups, high variability from human to human leads to unpredictable outcomes (Rashkin et al., 2023).

**Other benchmarks.** LiveCodeBench (Jain et al., 2024) also regularly releases new questions and makes use of ground-truth judging. However, it is limited to only coding tasks. The extensive Omni-MATH benchmark Gao & Liu (2024) encompasses numerous math competitions, although using LLM-as-a-judge grading potentially contributes to a degree of contamination or bias in some of the benchmark's scores; our completely objective correctness-based scoring avoids this concern. The SEAL Benchmark (Scale AI, 2024), uses private questions with expert human scorers, however, the benchmark currently only contains the following categories: Math, Coding, Instruction Following, and Spanish. In Srivastava et al. (2024), the authors modify the original MATH dataset (Hendrycks et al., 2021) by changing numbers in the problem setup. They find declines in model performance for some LLMs, including frontier ones. However, while such work can evaluate LLMs on data that is not in the pretraining set, the data still ends up being highly similar to the kind of data likely seen in the pretraining set. In addition, the hardness of the benchmark remains the same over time.

Finally, we discuss benchmarks that were the basis for tasks in `LiveBench`. In IFEval (Zhou et al., 2023b), the authors assess how good LLMs are at following instructions by adding one or more constraints in the instruction as to what the output should be. They limit the set of constraints to those in which it can provably be verified that the generation followed the constraints. In Big-Bench (Srivastava et al., 2022), a large number of tasks are aggregated into a single benchmark with the aim of being as comprehensive as possible. Big-Bench-Hard (Suzgun et al., 2022) investigates a subset of Big-Bench tasks that were particularly challenging for contemporaneous models as well as more complex prompting strategies for solving them.

## 5 CONCLUSIONS, LIMITATIONS, AND FUTURE WORK

In this work, we introduced LiveBench, an LLM benchmark designed to mitigate both test set contamination and the pitfalls of LLM judging and human crowdsourcing. LiveBench is the first benchmark that (1) contains frequently updated questions from new information sources, in which questions become harder over time, (2) scores answers automatically according to objective ground-truth values, without the use of LLM judges, and (3) contains a wide variety of challenging tasks, spanning math, coding, reasoning, language, instruction following, and data analysis. LiveBench contains questions that are based on recently released math competitions, arXiv papers, and datasets, and it contains harder, contamination-limited versions of previously released benchmarks. We released all questions, code, and model answers, and questions are added and updated on a monthly basis. We welcome community collaboration for expanding the benchmark tasks and models.

**Limitations and Future Work.** While we attempted to make `LiveBench` as diverse as possible, there are still additions from which it would benefit. For example, we hope to add non-English language tasks in the future. Furthermore, while ground truth scoring is beneficial in many ways, it still cannot be used for certain use cases, such as 'write a travel guide to Hawaii' in which it is hard to define a ground truth. Finally, while we attempted to make all tasks and categories fair for all models, there are still biases due to certain LLM families favoring certain prompt types. We plan to update the prompts (at the start and end of each question) in the future, as new prompt strategies are developed. Similarly, we plan to continue updating the `LiveBench` leaderboard as new LLMs are released.

## 6 REPRODUCIBILITY STATEMENT

Our work is fully reproducible: we open-source the leaderboard, all questions, all code to run API and open-source models, all model outputs for 40 models, and all code to score the models. In other words, every part of the project is available publicly: https://livebench.ai/. The only exception is that as the benchmark becomes more popular, we withhold releasing the new set of questions each month, so that there are always some questions that are private. These questions are then made public one month later. The readme in the above link gives instructions to download all parts of the project and to score new models.

## 7 ETHICS STATEMENT

Our paper introduces a new benchmark for LLMs, which contains frequently-updated questions from new information sources, scores answers according to objective ground-truth values, and contains a wide variety of tasks. We do not see any inherently negative broader societal impacts of our work.

Our hope is that our work will have a positive impact for both practitioners and researchers: by providing a new benchmark with frequently-updated questions, our work has the potential to both accelerate future research and enable more comprehensive and rigorous evaluations of existing and future models. Furthermore, we hope that the general framework of our benchmark – frequently-updated questions with new information sources – will catch on, mitigating the negative effects of contamination in future LLM evaluation and making LLM benchmarks more 'future-proof'.

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

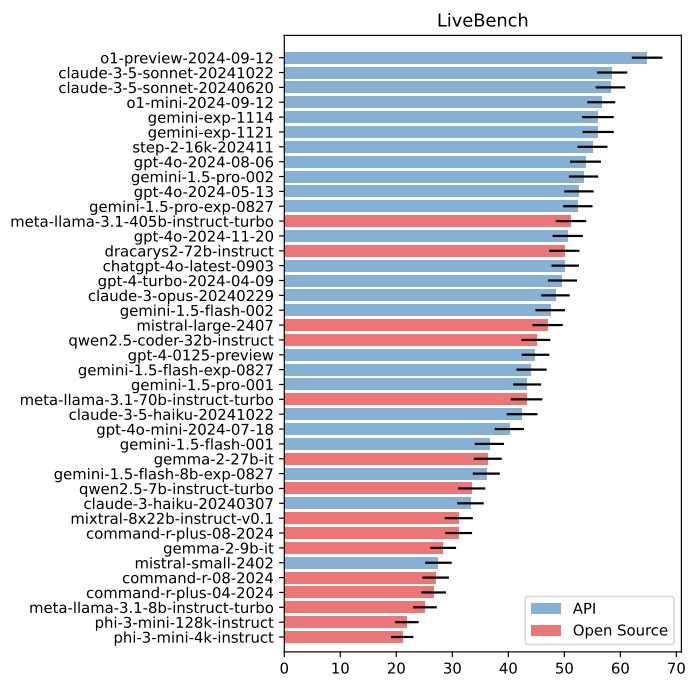

Figure 5: **Results on `LiveBench` for all models, showing 95% bootstrap confidence intervals.** This is the full version of Figure 1 (left).

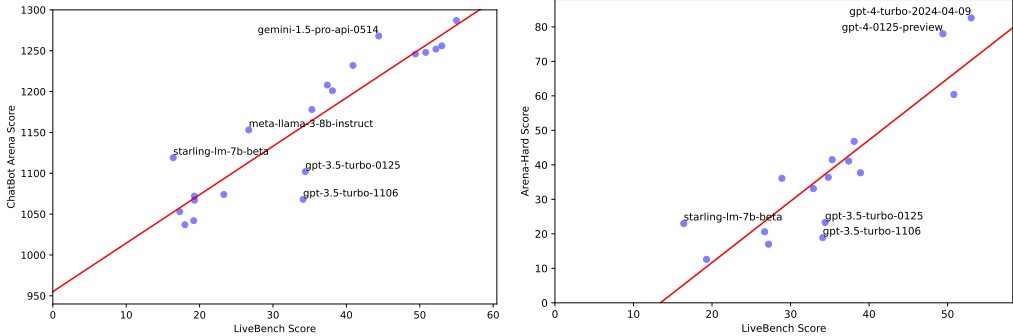

Figure 6: **The performance of models on different benchmarks, compared to a best-fit line.** We compare the different in relative performance of LLMs on `LiveBench` vs. ChatBot Arena, and `LiveBench` vs. Arena-Hard. We see that while many models are near the best-fit lines, a few are notable outliers, providing evidence that their output style may be noticeably better or worse than their ability to answer questions.

## A  ADDITIONAL DETAILS ABOUT LIVEBENCH EXPERIMENTS

In this section, we detail further descriptions about the `LiveBench` benchmark itself and our experiments.

We include further depictions of the comparisons of `LiveBench` to ChatBot Arena and Arena-Hard in Figure 6. We display the full results table for `LiveBench` in Table 2. We display the full bar plot for `LiveBench` in Figure 5.

We display the list of all verifiable instructions in Table 3.

We display a table with the citations for all models in Table 4.

Table 2: **LiveBench results across 40 models.** We output the results for each model on each main category, as well as each model's overall `LiveBench` score.

| Model | LiveBench Score | Coding | Data Analysis | Instruction Following | Language | Math | Reasoning |
|---|---|---|---|---|---|---|---|
| o1-preview-2024-09-12 | 64.7 | 50.8 | **64.0** | 74.6 | **68.7** | **62.9** | 67.4 |
| claude-3-5-sonnet-20241022 | 58.5 | **67.1** | 52.8 | 69.3 | 53.8 | 51.3 | 56.7 |
| claude-3-5-sonnet-20240620 | 58.2 | 60.8 | 56.7 | 68.0 | 53.2 | 53.3 | 57.2 |
| o1-mini-2024-09-12 | 56.7 | 48.1 | 54.1 | 65.4 | 40.9 | 59.2 | **72.3** |
| gemini-exp-1114 | 56.0 | 52.4 | 57.5 | 77.1 | 38.7 | 54.9 | 55.7 |
| gemini-exp-1121 | 56.0 | 50.4 | 57.0 | **80.1** | 40.0 | 62.8 | 45.8 |
| step-2-16k-202411 | 55.1 | 46.9 | 54.9 | 79.9 | 44.5 | 48.9 | 55.5 |
| gpt-4o-2024-08-06 | 53.8 | 51.4 | 52.9 | 68.6 | 47.6 | 48.2 | 53.9 |
| gemini-1.5-pro-002 | 53.4 | 48.8 | 52.3 | 70.8 | 43.3 | 57.4 | 47.9 |
| gpt-4o-2024-05-13 | 52.6 | 49.4 | 52.4 | 68.2 | 50.0 | 46.0 | 49.6 |
| gemini-1.5-pro-exp-0827 | 52.4 | 40.9 | 50.8 | 69.3 | 46.1 | 56.1 | 50.9 |
| meta-llama-3.1-405b-instruct-turbo | 51.1 | 43.8 | 53.5 | 72.8 | 43.2 | 40.5 | 52.8 |
| gpt-4o-2024-11-20 | 50.6 | 46.1 | 47.2 | 64.9 | 47.4 | 42.5 | 55.7 |
| dracarys2-72b-instruct | 50.1 | 56.6 | 49.1 | 65.2 | 33.5 | 50.6 | 45.8 |
| chatgpt-4o-latest-0903 | 50.1 | 47.4 | 48.7 | 66.4 | 45.3 | 42.1 | 50.5 |
| gpt-4-turbo-2024-04-09 | 49.6 | 49.0 | 51.3 | 60.8 | 44.3 | 42.7 | 49.5 |
| claude-3-opus-20240229 | 48.4 | 38.6 | 54.3 | 63.9 | 50.4 | 43.4 | 39.8 |
| gemini-1.5-flash-002 | 47.6 | 41.9 | 44.2 | 78.8 | 27.9 | 47.2 | 45.7 |
| mistral-large-2407 | 47.0 | 47.1 | 46.6 | 63.7 | 39.5 | 43.7 | 41.6 |
| qwen2.5-coder-32b-instruct | 45.0 | 56.8 | 43.4 | 58.7 | 23.2 | 45.9 | 42.1 |
| gpt-4-0125-preview | 44.8 | 41.8 | 54.1 | 57.2 | 39.2 | 33.4 | 43.2 |
| gemini-1.5-flash-exp-0827 | 44.1 | 40.6 | 47.9 | 73.0 | 29.6 | 28.9 | 44.5 |
| gemini-1.5-pro-001 | 43.3 | 32.3 | 52.8 | 60.2 | 40.4 | 36.9 | 37.0 |
| meta-llama-3.1-70b-instruct-turbo | 43.2 | 32.7 | 50.3 | 69.7 | 34.3 | 34.4 | 37.9 |
| claude-3-5-haiku-20241022 | 42.4 | 51.4 | 42.4 | 61.9 | 35.4 | 35.5 | 28.1 |
| gpt-4o-mini-2024-07-18 | 40.2 | 43.2 | 44.5 | 56.8 | 28.6 | 35.6 | 32.8 |
| gemini-1.5-flash-001 | 36.6 | 34.3 | 44.0 | 52.6 | 31.6 | 32.3 | 24.9 |
| gemma-2-27b-it | 36.4 | 35.9 | 43.6 | 58.1 | 32.6 | 26.2 | 22.1 |
| gemini-1.5-flash-8b-exp-0827 | 36.1 | 28.7 | 35.3 | 70.0 | 20.8 | 27.8 | 33.8 |
| qwen2.5-7b-instruct-turbo | 33.4 | 37.9 | 32.8 | 51.0 | 14.6 | 38.2 | 26.2 |
| claude-3-haiku-20240307 | 33.2 | 24.5 | 41.5 | 55.3 | 29.1 | 22.9 | 25.9 |
| mixtral-8x22b-instruct-v0.1 | 31.2 | 32.0 | 31.7 | 52.3 | 21.8 | 24.5 | 24.9 |
| command-r-plus-08-2024 | 31.1 | 19.5 | 35.9 | 57.6 | 29.7 | 19.3 | 24.8 |
| gemma-2-9b-it | 28.3 | 22.5 | 35.1 | 52.6 | 25.5 | 19.5 | 14.5 |
| mistral-small-2402 | 27.5 | 21.2 | 31.9 | 56.4 | 18.9 | 18.5 | 17.9 |
| command-r-08-2024 | 27.0 | 17.9 | 31.3 | 55.6 | 16.7 | 19.5 | 21.0 |
| command-r-plus-04-2024 | 26.6 | 19.5 | 24.6 | 59.5 | 19.7 | 16.8 | 19.8 |
| meta-llama-3.1-8b-instruct-turbo | 25.0 | 19.7 | 32.2 | 51.5 | 17.9 | 16.6 | 12.2 |
| phi-3-mini-128k-instruct | 21.9 | 15.0 | 34.0 | 39.1 | 9.2 | 14.6 | 19.6 |
| phi-3-mini-4k-instruct | 21.1 | 15.0 | 29.5 | 36.4 | 8.6 | 15.0 | 22.2 |

Table 3: The list of 25 instructions used in (Zhou et al., 2023a), and the 16 that are both 'real-world' and automatically verifiable, which we used in `LiveBench`. Descriptions are from (Zhou et al., 2023a).

| Instruction Group | Instruction | Description | In IFEval | In LiveBench |
|---|---|---|---|---|
| Keywords | Include Keywords | Include keywords {keyword1}, {keyword2} in your response | ✓ | ✓ |
| Keywords | Keyword Frequency | In your response, the word word should appear {N} times. | ✓ | |
| Keywords | Forbidden Words | Do not include keywords {forbidden words} in the response. | ✓ | ✓ |
| Keywords | Letter Frequency | In your response, the letter {letter} should appear {N} times. | ✓ | |
| Language | Response Language | Your ENTIRE response should be in {language}, no other language is allowed. | ✓ | |
| Length Constraints | Number Paragraphs | Your response should contain {N} paragraphs. You separate paragraphs using the markdown divider: * * * | ✓ | ✓ |
| Length Constraints | Number Words | Answer with at least / around / at most {N} words. | ✓ | ✓ |
| Length Constraints | Number Sentences | Answer with at least / around / at most {N} sentences. | ✓ | ✓ |
| Length Constraints | Number Paragraphs + First Word in i-th Paragraph | There should be {N} paragraphs. Paragraphs and only paragraphs are separated with each other by two line breaks. The {i}-th paragraph must start with word {first_word}. | ✓ | ✓ |
| Detectable Content | Postscript | At the end of your response, please explicitly add a postscript starting with {postscript marker} | ✓ | ✓ |
| Detectable Content | Number Placeholder | The response must contain at least {N} placeholders represented by square brackets, such as [address]. | ✓ | |
| Detectable Format | Number Bullets | Your answer must contain exactly {N} bullet points. Use the markdown bullet points such as: * This is a point. | ✓ | ✓ |
| Detectable Format | Title | Your answer must contain a title, wrapped in double angular brackets, such as <<poem of joy>>. | ✓ | ✓ |
| Detectable Format | Choose From | Answer with one of the following options: {options} | ✓ | |
| Detectable Format | Minimum Number Highlighted Section | Highlight at least {N} sections in your answer with markdown, i.e. *highlighted section* | ✓ | |
| Detectable Format | Multiple Sections | Your response must have {N} sections. Mark the beginning of each section with {section_splitter} X. | ✓ | ✓ |
| Detectable Format | JSON Format | Entire output should be wrapped in JSON format. | ✓ | ✓ |
| Combination | Repeat Prompt | First, repeat the request without change, then give your answer (do not say anything before repeating the request; the request you need to repeat does not include this sentence) | ✓ | ✓ |
| Combination | Two Responses | Give two different responses. Responses and only responses should be separated by 6 asterisk symbols: ******. | ✓ | ✓ |
| Change Cases | All Uppercase | Your entire response should be in English, capital letters only. | ✓ | |
| Change Cases | All Lowercase | Your entire response should be in English, and in all lowercase letters. No capital letters are allowed. | ✓ | |
| Change Cases | Frequency of All-capital Words | In your response, words with all capital letters should appear at least / around / at most {N} times. | ✓ | |
| Start with / End with | End Checker | Finish your response with this exact phrase {end_phrase}. No other words should follow this phrase. | ✓ | ✓ |
| Start with / End with | Quotation | Wrap your entire response with double quotation marks. | ✓ | ✓ |
| Punctuation | No Commas | In your entire response, refrain from the use of any commas. | ✓ | |

Table 4: List of models evaluated (across all LiveBench versions) and their respective citations.

| Model Name | Citation |
|---|---|
| chatgpt-4o-latest-0903 | (Hurst et al., 2024) |
| claude-3-5-haiku-20241022 | https://www.anthropic.com/claude/haiku |
| claude-3-5-sonnet-20240620 | https://www.anthropic.com/news/claude-3-5-sonnet |
| claude-3-5-sonnet-20241022 | https://www.anthropic.com/news/claude-3-5-sonnet |
| claude-3-haiku-20240307 | (Anthropic, 2024) |
| claude-3-opus-20240229 | (Anthropic, 2024) |
| claude-3-sonnet-20240229 | (Anthropic, 2024) |
| command-r | (Cohere, 2024) |
| command-r-08-2024 | (Cohere, 2024) |
| command-r-plus | (Cohere, 2024) |
| command-r-plus-08-2024 | (Cohere, 2024) |
| deepseek-coder-v2 | (DeepSeek-AI et al., 2024) |
| deepseek-coder-v2-lite-instruct | (DeepSeek-AI et al., 2024) |
| deepseek-v2-lite-chat | (DeepSeek-AI et al., 2024) |
| deepseek-v2.5 | (DeepSeek-AI et al., 2024) |
| dracarys-72b-instruct | https://huggingface.co/abacusai/Dracarys-72B-Instruct |
| dracarys-llama-3.1-70b-instruct | https://huggingface.co/abacusai/Dracarys-Llama-3.1-70B-Instruct |
| dracarys2-72b-instruct | https://huggingface.co/abacusai/Dracarys2-72B-Instruct |
| dracarys2-llama-3.1-70b-instruct | https://huggingface.co/abacusai/Dracarys2-Llama-3.1-70B-Instruct |
| gemini-1.5-flash-002 | (Reid et al., 2024) |
| gemini-1.5-flash-8b-exp-0827 | (Reid et al., 2024) |
| gemini-1.5-flash-api-0514 | (Reid et al., 2024) |
| gemini-1.5-flash-exp-0827 | (Reid et al., 2024) |
| gemini-1.5-pro-002 | (Reid et al., 2024) |
| gemini-1.5-pro-api-0514 | (Reid et al., 2024) |
| gemini-1.5-pro-exp-0801 | (Reid et al., 2024) |
| gemini-1.5-pro-exp-0827 | (Reid et al., 2024) |
| gemini-exp-1114 | https://ai.google.dev/gemini-api/docs/models/experimental-models |
| gemini-exp-1121 | https://ai.google.dev/gemini-api/docs/models/experimental-models |
| gemma-1.1-7b-it | (Team, 2024a) |
| gemma-2-27b-it | (Team et al., 2024) |
| gemma-2-2b | (Team et al., 2024) |
| gemma-2-9b-it | (Team et al., 2024) |
| gpt-3.5-turbo-0125 | (Brown et al., 2020) |
| gpt-3.5-turbo-1106 | (Brown et al., 2020) |
| gpt-4-0125-preview | (OpenAI, 2023) |
| gpt-4-0613 | (OpenAI, 2023) |
| gpt-4-1106-preview | (OpenAI, 2023) |
| gpt-4-turbo-2024-04-09 | (OpenAI, 2023) |
| gpt-4o-2024-05-13 | (Hurst et al., 2024) |
| gpt-4o-2024-08-06 | (Hurst et al., 2024) |
| gpt-4o-2024-11-20 | (Hurst et al., 2024) |
| gpt-4o-mini-2024-07-18 | (Hurst et al., 2024) |
| grok-2 | https://x.ai/blog/grok-2 |
| grok-2-mini | https://x.ai/blog/grok-2 |
| llama-2-7b-chat-hf | (Touvron et al., 2023) |
| llama-3.1-nemotron-70b-instruct | (Adler et al., 2024) |
| meta-llama-3-70b-instruct | (Meta, 2024) |
| meta-llama-3-8b-instruct | (Meta, 2024) |
| meta-llama-3.1-405b-instruct-turbo | (Dubey et al., 2024) |
| meta-llama-3.1-70b-instruct-turbo | (Dubey et al., 2024) |
| meta-llama-3.1-8b-instruct-turbo | (Dubey et al., 2024) |
| mistral-7b-instruct-v0.2 | (Jiang et al., 2023) |
| mistral-7b-instruct-v0.3 | (Jiang et al., 2023) |
| mistral-large-2402 | (Jiang et al., 2023) |
| mistral-large-2407 | (Jiang et al., 2023) |
| mistral-small-2402 | (Jiang et al., 2023) |
| mixtral-8x22b-instruct-v0.1 | (Jiang et al., 2023) |
| mixtral-8x7b-instruct-v0.1 | (Jiang et al., 2023) |
| o1-mini-2024-09-12 | https://openai.com/index/openai-o1-system-card/ |
| o1-preview-2024-09-12 | https://openai.com/index/openai-o1-system-card/ |
| open-mistral-nemo | (Jiang et al., 2023) |
| phi-3-medium-128k-instruct | (Abdin et al., 2024) |
| phi-3-mini-128k-instruct | (Abdin et al., 2024) |
| phi-3-small-128k-instruct | (Abdin et al., 2024) |
| phi-3.5-mini-instruct | (Abdin et al., 2024) |
| phi-3.5-moe-instruct | (Abdin et al., 2024) |
| qwen1.5-0.5b-chat | (Bai et al., 2023) |
| qwen1.5-72b-chat | (Bai et al., 2023) |
| qwen1.5-110b-chat | (Bai et al., 2023) |
| qwen1.5-7b-chat | (Bai et al., 2023) |
| qwen2-0.5b-instruct | (Yang et al., 2024) |
| qwen2-1.5b-instruct | (Yang et al., 2024) |
| qwen2-72b-instruct | (Yang et al., 2024) |
| qwen2-7b-instruct | (Yang et al., 2024) |
| qwen2.5-72b-instruct | (Team, 2024b) |
| qwen2.5-7b-instruct-turbo | (Team, 2024b) |
| starling-lm-7b-beta | (Zhu et al., 2023) |
| step-2-16k-202411 | https://www.stepfun.com/#step2 |
| vicuna-7b-v1.5 | (Chiang et al., 2023) |
| vicuna-7b-v1.5-16k | (Chiang et al., 2023) |
| yi-6b-chat | https://huggingface.co/01-ai/Yi-6B |
| zephyr-7b-alpha | (Tunstall et al., 2023) |
| zephyr-7b-beta | (Tunstall et al., 2023) |

Table 5: Pearson correlation coefficient and std. error for each category compared to the overall average `LiveBench` score, computed using data from all 40 models.

| Category | Correlation | Std Error |
|---|---|---|
| math | 0.9477 | 0.0643 |
| reasoning | 0.9439 | 0.0709 |
| data_analysis | 0.9315 | 0.0489 |
| coding | 0.8970 | 0.0840 |
| language | 0.8970 | 0.0831 |
| instruction_following | 0.8174 | 0.0811 |

Table 6: Pearson correlation coefficient and std. error for each task compared to the overall average `LiveBench` score, computed using data from all 40 models.

| Category | Correlation | Std Error |
|---|---|---|
| web_of_lies_v2 | 0.9136 | 0.1454 |
| math_comp | 0.9035 | 0.0986 |
| tablejoin | 0.8984 | 0.0961 |
| zebra_puzzle | 0.8853 | 0.0935 |
| coding_completion | 0.8623 | 0.1212 |
| cta | 0.8556 | 0.0486 |
| plot_unscrambling | 0.8553 | 0.0881 |
| LCB_generation | 0.8550 | 0.0816 |
| connections | 0.8368 | 0.1188 |
| AMPS_Hard | 0.8245 | 0.1357 |
| olympiad | 0.8192 | 0.1171 |
| summarize | 0.8044 | 0.0904 |
| paraphrase | 0.7884 | 0.0953 |
| simplify | 0.7765 | 0.0829 |
| typos | 0.7722 | 0.1474 |
| story_generation | 0.7443 | 0.1018 |
| spatial | 0.7294 | 0.0968 |
| tablereformat | 0.6840 | 0.1058 |

### A.1 DETAILS FROM CORRELATION ANALYSES

Here, we provide more details from Section 3.2. First, we present the Pearson correlation coefficient and std. error for each category (Table 5) and task (Table 6) compared to the overall average `LiveBench` score, computed using data from all 40 models. This is a supplement to Figure 2. In Table 7, we compute the relative best and worst task for each model, specifically, the tasks with the highest and lowest residuals of the best fit line vs. overall `LiveBench` performance. In other words, we compute the task that each model most outperforms and underperforms on, relative to a theoretical model with the same overall performance but has balanced performance across each task.

### A.2 DETAILS FROM ABLATION STUDIES

In this section, we ive the details for the ablation study described in Section 3.3. For hard math and reasoning questions, if an LLM struggles to answer the question, then will it also struggle to determine whether or not a given answer to that question is correct? There are some classes of problems for which the answer is surely 'no': any problems that are hard to solve by frontier LLMs, yet easy to check whether an answer is correct or not, such as NP-Hard problems. Another exception is that if an LLM judge is given access to the ground truth, then it will (of course) be able to judge whether or not answers are correct. The tasks in our original experiments (AMC, AIME, and Zebra puzzles) may not fit the class of exceptions. By way of a preliminary study of the above suggestion (that LLM judges cannot judge Zebra puzzles and AMC/AIME questions that they cannot solve), we run an experimental test. We use a judge prompt based on the MT-Bench judge prompt, which is duplicated below.

Table 7: Relative best and worst task for each model, computed as the tasks with the highest and lowest residuals of the best fit line vs. overall `LiveBench` performance, for each model.

| Model | Best | Worst |
|---|---|---|
| o1-preview-2024-09-12 | connections | coding_completion |
| claude-3-5-sonnet-20241022 | coding_completion | web_of_lies_v2 |
| claude-3-5-sonnet-20240620 | olympiad | math_comp |
| o1-mini-2024-09-12 | zebra_puzzle | coding_completion |
| gemini-exp-1114 | AMPS_Hard | typos |
| gemini-exp-1121 | story_generation | web_of_lies_v2 |
| step-2-16k-202411 | paraphrase | LCB_generation |
| gpt-4o-2024-08-06 | spatial | web_of_lies_v2 |
| gemini-1.5-pro-002 | olympiad | web_of_lies_v2 |
| gpt-4o-2024-05-13 | typos | tablejoin |
| gemini-1.5-pro-exp-0827 | olympiad | LCB_generation |
| meta-llama-3.1-405b-instruct-turbo | web_of_lies_v2 | connections |
| gpt-4o-2024-11-20 | typos | olympiad |
| dracarys2-72b-instruct | coding_completion | typos |
| chatgpt-4o-latest-0903 | spatial | olympiad |
| gpt-4-turbo-2024-04-09 | connections | simplify |
| claude-3-opus-20240229 | typos | web_of_lies_v2 |
| gemini-1.5-flash-002 | summarize | typos |
| mistral-large-2407 | olympiad | tablejoin |
| qwen2.5-coder-32b-instruct | LCB_generation | typos |
| gpt-4-0125-preview | tablereformat | olympiad |
| gemini-1.5-flash-exp-0827 | web_of_lies_v2 | AMPS_Hard |
| gemini-1.5-pro-001 | typos | web_of_lies_v2 |
| meta-llama-3.1-70b-instruct-turbo | tablereformat | zebra_puzzle |
| claude-3-5-haiku-20241022 | coding_completion | web_of_lies_v2 |
| gpt-4o-mini-2024-07-18 | AMPS_Hard | olympiad |
| gemini-1.5-flash-001 | typos | web_of_lies_v2 |
| gemma-2-27b-it | tablereformat | web_of_lies_v2 |
| gemini-1.5-flash-8b-exp-0827 | paraphrase | plot_unscrambling |
| qwen2.5-7b-instruct-turbo | AMPS_Hard | typos |
| claude-3-haiku-20240307 | typos | web_of_lies_v2 |
| mixtral-8x22b-instruct-v0.1 | coding_completion | tablereformat |
| command-r-plus-08-2024 | typos | coding_completion |
| gemma-2-9b-it | typos | spatial |
| mistral-small-2402 | web_of_lies_v2 | zebra_puzzle |
| command-r-08-2024 | paraphrase | AMPS_Hard |
| command-r-plus-04-2024 | simplify | tablereformat |
| meta-llama-3.1-8b-instruct-turbo | typos | spatial |
| phi-3-mini-128k-instruct | spatial | story_generation |
| phi-3-mini-4k-instruct | spatial | simplify |

[Instruction] Please act as an impartial judge and evaluate the quality of the response provided by an AI assistant to the user question displayed below. Your evaluation should consider correctness alone. Identify and correct any mistakes. Be as objective as possible. After providing your explanation, you must rate the response as either 1 (correct) or 0 (incorrect) by strictly following this format: "[[rating]]", for example: "Rating: [[1]]" [Question] question [The Start of Assistant's Answer] answer [The End of Assistant's Answer]

We use `gpt-4-turbo-2024-04-09` as the judge. We judge the model outputs of both `gpt-4-turbo-2024-04-09` and `claude-3-opus-20240229`.

See Table 8 and Table 9. We find that the error rate for all tasks is far above a reasonable value, indicating that LLM judges are not appropriate for challenging math and logic tasks. However, we

Table 8: **LLM judges cannot accurately evaluate challenging math and reasoning questions.** Error rate of LLM-as-a-judge scoring on challenging math (AMC, AIME, SMC) and reasoning (Zebra puzzles) tasks. On all tasks, the error rate is surprisingly high, showing that LLMs are not reliable judges for these tasks.

| Model | Judge | AMC12 2024 | AIME 2024 | SMC 2023 | Zebra Puzzles |
|---|---|---|---|---|---|
| GPT-4-Turbo | GPT-4-Turbo | 0.380 | 0.214 | 0.353 | 0.420 |
| Claude-3-Opus | GPT-4-Turbo | 0.388 | 0.103 | 0.294 | 0.460 |

Table 9: Model Performance on math and reasoning tasks with both ground-truth (GT) or LLM judging (LLM-Jdg.)

| | AMC12 2024 | | AIME 2024 | | SMC 2023 | | Zebra Puzzles | |
|---|---|---|---|---|---|---|---|---|
| | GT | LLM-Jdg. | GT | LLM-Jdg. | GT | LLM-Jdg. | GT | LLM-Jdg. |
| GPT-4-Turbo | 54 | 64.000 | 13.793 | 35.714 | 70.588 | 58.824 | 38 | 68 |
| Claude-3-Opus | 56 | 42.857 | 6.897 | 17.241 | 58.824 | 52.941 | 34 | 52 |

note that there may be other experimental setups which could change the result, such as using a more detailed prompt that is tailored to the task of judging hard math and reasoning problems.

## A.3 DETAILED DESCRIPTION OF LIVEBENCH CATEGORIES

In this section, we describe the categories and tasks of `LiveBench` and the grading methods in more detail.

### A.3.1 MATH CATEGORY

Evaluating the mathematical abilities of LLMs has been one of the cornerstones of recent research in LLMs, featuring prominently in many releases and reports (Reid et al., 2024; OpenAI, 2023; Brown et al., 2020; Bubeck et al., 2023). Our benchmark includes math questions of three types: modified questions from recent high school math competitions, fill-in-the-blank questions from recent proof-based USAMO and IMO problems, and questions from our new, harder version of the AMPS dataset (Hendrycks et al., 2021).

**Math competitions.** Our first math category is based on expert human-designed math problems that offer a wider variety in terms of problem type and solution technique. We focus on high school math competition questions from English-speaking countries: AMC12, AIME, SMC, and USAMO, and also IMO, the international competition.

First, we include questions based on the *American Mathematics Competition 12 (AMC12)*, both AMC12A and AMC12B 2023, released on November 8, 2023 and November 14, 2023, respectively, and the Senior Mathematical Challenge (SMC) 2023, released on October 3, 2023. All three are challenging multiple-choice competitions for high school students in the USA (AMC) and UK (SMC) that build in difficulty, meant as the first step for high school students to qualify for their country's team for the International Mathematical Olympiad (IMO).

The questions test mathematical problem solving with arithmetic, algebra, counting, geometry, number theory, probability, and other secondary school math topics (Faires & Wells, 2022). We modify the questions by updating the prose of the questions that do not affect the answer, by rearranging the order of the multiple choice answers when applicable, and by asking for a different output format than the widely-used source website (https://artofproblemsolving.com/). An example of a problem of this type from the AMC12A 2023 problem set is below:

> **An example question from the Math Competitions task.**
> How many complex numbers satisfy the equation $z^5 = \overline{z}$, where $\overline{z}$ is the conjugate of the complex number $z$?  **(A)** $2$  **(B)** $3$  **(C)** $5$  **(D)** $6$  **(E)** $7$
> If you cannot determine the correct multiple-choice answer, take your best guess. Once you have your answer, please duplicate that letter five times in a single string. For example, if the answer is F, then write FFFFF.
>
> **Ground Truth:** EEEEE

Next, we include the *American Invitational Mathematics Examination (AIME)*, both AIME I and AIME II 2024, released on January 31, 2024 and February 7, 2024, respectively. These are prestigious and challenging tests given to those who rank in the top 5% of the AMC. Each question's answer is an integer from $000$ to $999$. An example of a problem of this type from the AIME I 2024 problem set is below:

> **An example question from the Math Competitions task.**
> Real numbers $x$ and $y$ with $x, y > 1$ satisfy $\log_x(y^x) = \log_y(x^{4y}) = 10$. What is the value of $xy$? Please think step by step, and then display the answer at the very end of your response. The answer is an integer consisting of exactly 3 digits (including leading zeros), ranging from 000 to 999, inclusive. For example, the answer might be 068 or 972. If you cannot determine the correct answer, take your best guess. Remember to have the three digits as the last part of the response.
>
> **Ground Truth:** 025

**Proof-based questions.**  We consider the *USA Math Olympiad (USAMO)* 2024 and *International Math Olympiad (IMO)* 2024 competitions, released on March 20, 2024 and July 22, 2024, respectively. These contests are primarily proof-based and non-trivial to evaluate in an automated way. One possibility is to use LLMs to evaluate the correctness of the natural language proof. However, we then have *no* formal guarantees on the correctness of the evaluation. Another possibility is to *auto-formalize* the proofs into a formal language such as Lean and then run a proof checker. However, while there have been notable recent improvements in auto-formalization, such a process still does not have formal guarantees on the correctness of the auto-formalization – and thus that of the evaluation. To tackle this, we formulate a novel task which can test the ability of an LLM in the context of proofs. Specifically, for a proof, we *mask* out a subset of the formulae in the proof. We then present the masked out formulae in a *scrambled* order to the LLM and ask it to *reinsert* the formulae in the correct positions. Such a task tests the mathematical, deductive, and instruction following abilities of the LLM. In particular, if the LLM is strong enough to generate the correct proof for a question, then one would expect it to also solve the far easier task of completing a proof which has some missing formulae – especially if the formulae are already given to it in a scrambled order. Note that this also allows us to easily control the level of difficulty of the question by changing the number of formulae that we mask.

We generate 3 hardness variants for each problem, masking out 10%, 50% and 80% of the equations in the proof. We evaluate by computing the edit distance between the ground truth ranking order and the model predicted ranking order. [NB : in preliminary testing we also evaluated using the accuracy metric and the model rankings remained nearly the same]. Models perform worse on IMO compared to USAMO, in line with expectations. We also looked at the performance as separated by question hardness. The scores are greatly affected by question hardness going from as high as 96.8 for the easiest questions (10% masked out, GPT-4o) to as low as 36 for the hardest (80% masked out). The full results are in Table 10 and Table 11.[1]

**Synthetically generated math questions.**  Finally, we release synthetic generated math questions. This technique is inspired from math question generation used to create the MATH and AMPS datasets (Hendrycks et al., 2021). In particular, we randomly generate a math problem of one of several types, such as taking the derivative or integral of a function, completing the square, or factoring a polynomial. We generate questions by drawing random primitives, using a larger (and therefore more challenging) distribution than AMPS. Note that, for problem types such as integration,

---

[1]Note that these experiments were run in June 2024, before models such as claude-3.5-sonnet were released.

Table 10: IMO/USAMO results for each of 34 models across all hardness levels.

| Model | IMO | USAMO | Avg. |
|---|---|---|---|
| gpt-4o-2024-05-13 | 60.24 | 67.47 | 63.85 |
| gpt-4-1106-preview | 58.16 | 67.17 | 62.66 |
| claude-3-opus-20240229 | 52.56 | 63.66 | 58.11 |
| gpt-4-turbo-2024-04-09 | 50.96 | 64.80 | 57.88 |
| gemini-1.5-pro-latest | 52.11 | 59.15 | 55.63 |
| gpt-4-0125-preview | 43.04 | 60.66 | 51.85 |
| Meta-Llama-3-70B-Instruct | 43.24 | 59.55 | 51.40 |
| claude-3-sonnet-20240229 | 44.78 | 52.97 | 48.87 |
| command-r-plus | 48.33 | 44.55 | 46.44 |
| gpt-3.5-turbo-1106 | 40.37 | 49.65 | 45.01 |
| mistral-large-2402 | 38.65 | 50.41 | 44.53 |
| claude-3-haiku-20240307 | 41.51 | 47.31 | 44.41 |
| gpt-3.5-turbo-0125 | 38.44 | 47.17 | 42.80 |
| Qwen1.5-72B-Chat | 34.35 | 48.47 | 41.41 |
| Mixtral-8x22B-Instruct-v0.1 | 33.00 | 48.62 | 40.81 |
| mistral-small-2402 | 34.51 | 44.78 | 39.64 |
| Meta-Llama-3-8B-Instruct | 36.05 | 36.59 | 36.32 |
| Qwen1.5-110B-Chat | 23.93 | 46.78 | 35.35 |
| Mistral-7B-Instruct-v0.2 | 36.00 | 34.31 | 35.15 |
| command-r | 31.36 | 29.38 | 30.37 |
| Phi-3-mini-128k-instruct | 25.84 | 33.54 | 29.69 |
| Mixtral-8x7B-Instruct-v0.1 | 26.52 | 32.50 | 29.51 |
| Phi-3-mini-4k-instruct | 26.60 | 30.33 | 28.46 |
| Qwen1.5-7B-Chat | 22.10 | 31.84 | 26.97 |
| Starling-LM-7B-beta | 14.99 | 28.70 | 21.84 |
| zephyr-7b-alpha | 25.99 | 16.43 | 21.21 |
| vicuna-7b-v1.5-16k | 23.14 | 16.69 | 19.91 |
| Yi-6B-Chat | 18.17 | 20.05 | 19.11 |
| zephyr-7b-beta | 9.57 | 22.57 | 16.07 |
| Llama-2-7b-chat-hf | 20.00 | 11.53 | 15.77 |
| Qwen1.5-4B-Chat | 11.90 | 16.78 | 14.34 |
| vicuna-7b-v1.5 | 16.19 | 9.87 | 13.03 |
| Qwen1.5-0.5B-Chat | 9.27 | 10.61 | 9.94 |
| Qwen1.5-1.8B-Chat | 0.98 | 9.13 | 5.06 |

Table 11: IMO/USAMO results for each hardness level across 34 models.

| Hardness level | Avg. | IMO | USAMO |
|---|---|---|---|
| Easy | 57.48 | 54.68 | 60.27 |
| Medium | 29.60 | 25.79 | 33.41 |
| Hard | 19.11 | 15.96 | 22.25 |

this simple technique of drawing a random function and taking its derivative results in a wide variety of integration problems of varying difficulty. For example, problem solutions may involve applying the chain rule, the product/quotient rule, trigonometric identities, or use a change of variables. In order to extract the answer, we ask the model to use the same 'latex boxed answer' technique as in the MATH dataset (Hendrycks et al., 2021). We judge the correctness of answers as in the EleutherAI Eval Harness (Gao et al., 2021) using Sympy (Meurer et al., 2017) where we check for semantic as well as numerical equivalence of mathematical expressions. An example of a integral problem is as follows:

> **An example question from the AMPS Hard task.**
> Find an indefinite integral (which can vary by a constant) of the following function: $5\sec^2(5x+1) - 8\sin(7-8x)$. Please put your final answer in a $boxed\{\}$.
>
> **Ground Truth:** $-\sin(7)\sin(8x) - \cos(7)\cos(8x) + \tan(5x+1)$

### A.3.2 CODING CATEGORY

The coding ability of LLMs is one of the most widely studied and sought-after skills for LLMs (Mnih et al., 2015; Jain et al., 2024; Li et al., 2023a). We include two coding tasks in `LiveBench`: a modified version of the code generation task from LiveCodeBench (Jain et al., 2024), and a novel code completion task combining LiveCodeBench problems with partial solutions collected from GitHub sources. Examples of questions from the Coding tasks can be found here.

**Code generation.**  In the `LCB Generation` task, we assess a model's ability to parse a competition coding question statement and write a correct answer. LiveCodeBench (Jain et al., 2024) included several tasks to assess the coding capabilities of large language models. We have taken 78 randomly selected problems from the April 2024 release of LiveCodeBench, selecting only problems released in or after November 2023. The problems are competition programming problems from LeetCode (Team, 2015) and AtCoder(Team, 2012), defined with a textual description and solved by writing full programs in Python 3 code.

These problems are presented as in LiveCodeBench's Code Generation task, with minor prompting differences and with only one chance at generating a correct solution per question, per model. We report pass@1, a metric which describes the proportion of questions that a given model solved completely (a solution is considered correct if and only if it passes all public and private test cases).

**Code completion.**  In this task, we assess the ability of the model to successfully complete a partially provided solution to a competition coding question statement. The setup is similar to the Code Generation task above, but a partial (correct) solution is provided in the prompt and the model is instructed to complete it to solve the question. We use LeetCode easy, medium, and hard problems from LiveCodeBench's (Jain et al., 2024) April 2024 release, combined with matching solutions from `https://github.com/kamyu104/LeetCode-Solutions`, omitting the last 15% of each medium/hard solution and 30-70% of each easy solution and asking the LLM to complete the solution. As with Code Generation, we report pass@1.

### A.3.3 REASONING CATEGORY

The reasoning abilities of large language models is another highly-benchmarked and analyzed skill of LLMs (Wei et al., 2022; Suzgun et al., 2023; Yao et al., 2024). In `LiveBench`, we include two reasoning tasks: a harder version of a task from Big-Bench Hard (Suzgun et al., 2023), and Zebra puzzles.

**Web of lies v2.**  Web of Lies is a task included in Big-Bench (bench authors, 2023) and Big-Bench Hard (Suzgun et al., 2023). The task is to evaluate the truth value of a random Boolean function expressed as a natural-language word problem. In particular, the LLM must evaluate $f_n(f_{n-1}(...f_1(x)...))$, where each $f_i$ is either negation or identity, and $x$ is True or False. We represent $x$ by the sentence: $X_0$ {tells the truth, lies}, and we represent $f_i$ by a sentence: $X_i$ says $X_{i-1}$ {tells the truth, lies}. The sentences can be presented in a random order for increased difficulty. For example, a simple $n = 2$ version is as follows: 'Ka says Yoland tells the truth. Yoland lies. Does Ka tell the truth?' Already by October 2022, LLMs achieved near 100% on this task, and furthermore, there are concerns that Big-Bench tasks leaked into the training data of GPT-4, despite using canary strings (OpenAI, 2023).

For `LiveBench`, we create a new, significantly harder version of Web of Lies. We make the task harder with a few additions: *(1)* adding different types of red herrings, *(2)* asking for the truth values of three people, instead of just one person, and *(3)* adding a simple additional deductive component. For *(1)*, we maintain a list of red herring names, so that the red herrings do not affect the logic of the answer while still potentially leading LLMs astray. For example, 'Fred says Kayla lies,' where Fred

is in the true 'web of lies', while Kayla may lead to a series of steps ending in a dead end. Overall, the number of total red herring sentences is drawn from a uniform distribution ranging from 0 to 19. For *(3)*, we simply assign each name to a location and give sentences of the form 'Devika is at the museum. The person at the museum says the person at the ice skating rink lies.' We find that this makes the task significantly harder for leading LLMs, even without shuffling the sentences into a random order.

---

**An example question from the Web of Lies v2 task.**
In this question, assume each person either always tells the truth or always lies. Tala is at the movie theater. The person at the restaurant says the person at the aquarium lies. Ayaan is at the aquarium. Ryan is at the botanical garden. The person at the park says the person at the art gallery lies. The person at the museum tells the truth. Zara is at the museum. Jake is at the art gallery. The person at the art gallery says the person at the theater lies. Beatriz is at the park. The person at the movie theater says the person at the train station lies. Nadia is at the campground. The person at the campground says the person at the art gallery tells the truth. The person at the theater lies. The person at the amusement park says the person at the aquarium tells the truth. Grace is at the restaurant. The person at the aquarium thinks their friend is lying. Nia is at the theater. Kehinde is at the train station. The person at the theater thinks their friend is lying. The person at the botanical garden says the person at the train station tells the truth. The person at the aquarium says the person at the campground tells the truth. The person at the aquarium saw a firetruck. The person at the train station says the person at the amusement park lies. Mateo is at the amusement park. Does the person at the train station tell the truth? Does the person at the amusement park tell the truth? Does the person at the aquarium tell the truth? Think step by step, and then put your answer in **bold** as a list of three words, yes or no (for example, **yes, no, yes**). If you don't know, guess.

**Ground Truth:** no, yes, yes

---

**Zebra puzzles.** The second reasoning task we include is Zebra puzzles. Zebra puzzles, also called Einstein's riddles or Einstein's puzzles, are a well-known (Jeremy, 2009) reasoning task that tests the ability of the model to follow a set of statements that set up constraints, and then logically deduce the requested information. The following is an example with three people and three attributes:

---

**An example question from the Zebra Puzzle task.**
There are 3 people standing in a line numbered 1 through 3 in a left to right order.
Each person has a set of attributes: Food, Nationality, Hobby.
The attributes have the following possible values:

- Food: nectarine, garlic, cucumber

- Nationality: chinese, japanese, thai

- Hobby: magic-tricks, filmmaking, puzzles

and exactly one person in the line has a given value for an attribute.
Given the following premises about the line of people:

- the person that likes garlic is on the far left

- the person who is thai is somewhere to the right of the person who likes magic-tricks

- the person who is chinese is somewhere between the person that likes cucumber and the person who likes puzzles

Answer the following question:
What is the hobby of the person who is thai? Return your answer as a single word, in the following format: ***X***, where X is the answer.

**Ground Truth:** filmmaking

---

We build on an existing repository for procedural generation of Zebra puzzles (quint t, 2023); the repository allows for randomizing the number of people, the number of attributes, and the set of constraint statements provided. For the attribute randomization, they are drawn from a set of 10 possible categories (such as Nationality, Food, Transport, Sport) and for each of these categories there are between 15 and 40 possible values to be taken. For the constraint statements, the implementation

allows for up to 20 'levels' of constraint in ascending order of intended difficulty. For example, level 1 could include a statement such as 'The person who likes garlic is on the left of the person who plays badminton' and a level 10 statement could be 'The person that watches zombie movies likes apples or the person that watches zombie movies likes drawing, but not both'. Higher levels also include lower level statements in their possible set of statements to draw from, but this set narrows progressively as the level increases from 12 to 20 by removing the possibility of having lower-level statements (starting with removing level 1, then removing level 2, etc).

The repository also includes a solver for the puzzles, which we use to ensure there is a (unique) solution to all of our generated puzzles.

Our modifications to the original repository primarily target the reduction of ambiguity in the statements (e.g. changing 'X is to the left of Y' to 'X is to the *immediate* left of Y'). For generation, we pick either 3 or 4 people with 50% probability, either 3 or 4 attributes with 50% probability, and we draw the levels from the integer interval [10, 20] with uniform probability. In preliminary testing, we found that larger puzzles proved exceedingly difficult for even the top performing LLMs.

**Spatial reasoning.** The final reasoning task is spatial reasoning questions. This set of 50 handwritten questions tests a model's ability to make deductions about intersections and orientations of common 2D and 3D shapes. Two example questions are below.

---

**Example question one from the Spatial Reasoning task.**
Suppose I have three spheres of radius 3 resting on a plane. Each sphere is tangent to the other two spheres. If I consider a new shape whose vertices are equal to the set of tangent points of the pairs of spheres, what is the new shape? Is it a square, tetrahedron, triangle, circle, line segment, or rhombus? Think step by step, and then put your answer in **bold** as a single phrase (for example, **circle**). If you don't know, guess.

**Ground Truth:** triangle

---

**Example question two from the Spatial Reasoning task.**
Suppose I have a regular heptagon, and I can make four straight cuts. Each cut cannot pass through any of the vertices of the heptagon. Also, exactly two of the cuts must be parallel. What is the maximum number of resulting pieces? Think step by step, and then put your answer in **bold** as a single integer (for example, **0**). If you don't know, guess.

**Ground Truth:** 10

---

### A.3.4 DATA ANALYSIS

LiveBench includes three practical tasks in which the LLM assists in data analysis or data science: column type annotation, table join prediction, and table reformatting. Each question makes use of a recent dataset from Kaggle or Socrata.

Owing to the limited output context lengths of the current generation of LLMs and the comparatively high per-token costs of generating responses, we upper bound the size of our tables with respect to cell length, column count and row count. Even with these limitations, we find that our tasks remain sufficiently challenging even for the current state-of-the-art models.

Example questions from the Data Analysis category can be lengthy, so examples can be viewed here.

**Column type annotation.** Consider a table $A$ with $t$ columns and $r$ rows. We denote each column $C \in A$ as a function which maps row indices to strings; i.e., for $0 \le i < t$, we have $C_i : \mathbb{N} \to \Sigma_*$, where $i$ is the column index. Let $L \subseteq \Sigma_*$ denote a label set; these are our column types to be annotated. Standard CTA assumes a fixed cardinality for this label set, indexed by a variable we call $j$. Given the above definitions, we define single-label $CTA \subset A \times L$ as a relation between tables and labels:

$$\forall C, \exists l_j \mid (C_i, l_j) \in CTA \tag{1}$$

We seek a generative method $M : \Sigma_* \to \Sigma_*$ that comes closest to satisfying the following properties:

$$M(\sigma, L) \in L, \forall C \in A, M(\sigma, L) \in CTA \qquad (2)$$

For further details on the task, please refer to Feuer et al. (2023). **Implementation details.** For each benchmark instance, we retrieve a random $A$ from our available pool of recent tables. We randomly and uniformly sample $C$ from $A$, use the actual column name of $A$ as our CTA ground-truth $L$, and retrieve $\sigma_1 \cdots \sigma_5$ column samples from $C$, with replacement, providing them as context for the LLM. **Metrics.** We report Accuracy @ 1 over all instances, accepting only case-insensitive exact string matches as correct answers.

**Table reformatting.** Given a table $A$ rendered according to a plaintext-readable and valid schema for storing tabular information $a_s$, we instruct the LLM to output the same table with the contents unchanged but the schema modified to a distinct plaintext-readable valid schema $b_s$. **Implementation details.** We use the popular library Pandas to perform all of our conversions to and from text strings. We allow the following formats for both input and output: "JSON", "JSONL", "Markdown", "CSV", "TSV", "HTML". As tabular conversion from JSON to Pandas is not standardized, we accept several variations. At inference time, we ingest the LLM response table directly into Pandas. **Metrics.** We report Accuracy @ 1 over all instances. An instance is accepted only if it passes all tests (we compare column count, row count, and exact match on row contents for each instance).

**Join-column prediction.** Given two tables $A$ and $B$, with columns $a_1, \ldots$ and $b_1, \ldots$ respectively, the *join-column prediction* task is to suggest a pair $(a_k, b_l)$ of columns such that the equality condition $a_k = b_l$ can be used to join the the tables in a way that matches with the provided ground-truth mapping $M : A \to B$. The mapping is usually partial injective: not every column in $B$ is mapped from $A$, not every column in $A$ is mapped to $B$. For further details, please refer to Yan & He (2020). **Implementation details.** We randomly sample columns with replacement from our entire collection of tables, generating a fixed column pool C. We retain half the rows of A to provide as context to the LLM. The remaining rows are used to generate a new table B. For each instance, we randomly sample columns from both the target table and the column pool and join them to B. We anonymize the column names in B, then pass both A and B to the LLM and ask it to return a valid join mapping M. **Metrics.** We report the F1 score over columns, with TPs scored as exact matches between ground truth and the LLM output, FPs scored as extraneous mappings, FNs scored as missing mappings, and incorrect mappings counting as FP + FN.

### A.3.5 INSTRUCTION FOLLOWING

An important ability of an LLM is its capability to follow instructions. To this end, we include instruction following questions in our benchmark, inspired by IFEval (Zhou et al., 2023a).

**Generating live prompts and instruction.** IFEval, or instruction-following evaluation for LLMs, contains verifiable instructions such as "write more than 300 words" or "Finish your response with this exact phrase: {end_phrase}." These instructions are then appended to prompts like "write a short blog about the a visit in Japan". We use this modular nature between the prompt and instruction to construct live prompts.

For our live source, we considered news articles from The Guardian; we are able to obtain 200 articles using their API[2]. Using the first $n$ sentences article text as the source text, we consider four different tasks using the text: paraphrase, summarize, simplify, and story generation. The exact prompts can be seen in Table 12. For the instructions, we use the code provided by Zhou et al. (2023a), making a few modifications such as increasing the max number of keywords from two to five. Additionally, we compose different instructions together by sampling from a uniform distribution from 2 to 5. However, since the instructions can be conflicting, we deconflict the instructions. This results in approximate normal distribution of the number of instructions per example with the majority of the containing two or three instructions. A full list of the instructions can be found in Appendix Table 3. To construct, the full prompt, containing the news article sentences, the prompt, and the instructions, we use the following meta prompt: "The following are the beginning sentences of a news article from the Guardian.\n———-\n{guardian article}\n———-\n{subtask prompt} {instructions}".

---

[2]https://open-platform.theguardian.com/

Table 12: The prompt for each subtask used in each of the four instruction following tasks.

| Subtask | Subtask Prompt |
|---|---|
| Paraphrase | Please paraphrase based on the sentences provided. |
| Summarize | Please summarize based on the sentences provided. |
| Simplify | Please explain in simpler terms what this text means. |
| Story Generation | Please generate a story based on the sentences provided. |

**Scoring.** To evaluate the model's performance on instruction following, we use a scoring method that considers two key factors: whether all instructions were correctly followed for a given prompt, i.e. Prompt-level accuracy, and what fraction of the individual instructions were properly handled, i.e. Instruction-level accuracy. The first component of the score checks if the model successfully followed every instruction in the prompt and assigns 1 or 0 if it missed any of the instructions. The second component looks at each individual instruction and checks whether it was properly followed or not. The final score is the average of these two components, scaled to lie between 0 and 1. A score of 1 represents perfect adherence to all instructions, while lower scores indicate varying degrees of failure in following the given instructions accurately.

Example questions from the Instruction Following category can be lengthy, so examples can be viewed here.

### A.3.6 LANGUAGE COMPREHENSION

Finally, we include multiple language comprehension tasks. These tasks assess the language model's ability to reason about language itself by, (1) completing word puzzles, (2) fixing misspellings while leaving other stylistic changes in place, and (3) reordering scrambled plots of unknown movies.

**Connections.** First we include the 'Connections' category[3]. Connections is a word puzzle category introduced by the New York Times (although similar ideas have existed previously). Sixteen words are provided in a random order; the objective of the game is to sort these into four sets of four words, such that each set has a 'connection' between them. Such connections could include the words belonging to a related category, e.g., 'kiwi, grape, pear, peach' (types of fruits); the words being anagrams, the words being homophones, or being words that finish a certain context, e.g., 'ant, drill, island, opal' being words that come after the word 'fire' to make a phrase. Due to the variety of possible connection types that can exist, the wider knowledge required to understand some connections, as well as some words potentially being 'red herrings' for connections, this task is challenging for LLMs – prior work (Todd et al., 2024) has comprehensively tested the task on the GPT family of models, as well as on sentence embedding models derived from, e.g., BERT (Devlin et al., 2018) and RoBERTa (Liu et al., 2019). The authors found that GPT-4 has an overall completion rate below 40% on the puzzles (when allowed multiple tries to get it correct), concluding that 'large language models in the GPT family are able to solve these puzzles with moderate reliability, indicating that the task is possible but remains a formidable challenge.' In our work, we assess the single-turn performance and test performance on a much larger set of models.

The original task provided for a number of 'retry' attempts in the event of an incorrect submission for a category. To fit into the framework of our benchmark we take the model's answer from a single turn; to ameliorate the increased difficulty of this setting, we use fewer words/groups for some questions. The split we use is 15 questions of eight words, 15 questions of twelve words and 20 questions of sixteen words. An example prompt is as follows:

---

[3]See https://www.nytimes.com/games/connections.

> **An example question from the `Connections` task.**
> You are given 8 words/phrases below. Find two groups of four items that share something in common. Here are a few examples of groups: bass, flounder, salmon, trout (all four are fish); ant, drill, island, opal (all four are two-word phrases that start with 'fire'); are, why, bee, queue (all four are homophones of letters); sea, sister, sin, wonder (all four are members of a septet). Categories will be more specific than e.g., '5-letter-words', 'names', or 'verbs'. There is exactly one solution. Think step-by-step, and then give your answer in **bold** as a list of the 8 items separated by commas, ordered by group (for example, **bass, founder, salmon, trout, ant, drill, island, opal**). If you don't know the answer, make your best guess. The items are: row, drift, curl, tide, current, press, fly, wave.
>
> **Ground Truth:** current, drift, tide, wave, curl, fly, press, row

The score for this task is the fraction of groups that the model outputs correctly.

**Typo corrections.** Next, we include details about the `Typos` task. The idea behind this task is inspired by the common use-case for LLMs where a user will ask the system to identify typos and misspellings in some written text. The challenge for the systems is to fix just the typos or misspellings, but to leave other aspects of the text unchanged. It is common for the LLM to impose its own writing style onto that of the input text, such as switching from British to US spellings or adding the serial comma, which may not be desirable.

To create the questions for this task, we take text from recent ArXiv abstracts. These abstracts may themselves start with misspellings and grammatical errors. Therefore, our first step is to manually pass over the abstracts and fix typos and grammar issues. Next, we assemble a list of common misspellings as found online. This is done so as to replicate common misspellings performed by humans, even though we synthetically generate the questions. Finally, for each question, we sample a probability $p \sim U(0.5, 0.7)$ of flipping correctly spelled words to misspelled words. We then use that probability to replace every correctly spelled word with a common misspelling with that probability $p$. This allows there to be variability in the difficulty of the problem included in the benchmark. In our first release, we include 50 questions. Finally, to score this problem, we merely ask whether the ground truth abstract is contained in the output provided by the LLM.

> **An example question from the `Typos` task.**
> Please output this exact text, with no changes at all except for fixing the misspellings. Please leave all other stylistic decisions like commas and US vs British spellings as in the original text.
> We introducehten consept of a $k$-token signed graph adn studdy some of its combinatorial and algebraical properties. We prove that twpo switching isomorphic signed graphs ahve switching isomorphic token graphs. Moreover, we sohw tyhat the Laplacian spectum of a balanced signed graph is contained in the Laplacian spectra of its $k$-token signed graph. Besides, we introduce and studdyther unbalance levle of a signed graph, which is a new parameter tyhat measures how far a signed graph is frome beng balanced. Moreover, we study the relation bewteen the frustration index anbdther unballance level of signed graphs adn their token signed graphs.
>
> **Ground Truth:** We introduce the concept of a $k$-token signed graph and study some of its combinatorial and algebraic properties. We prove that two switching isomorphic signed graphs have switching isomorphic token graphs. Moreover, we show that the Laplacian spectrum of a balanced signed graph is contained in the Laplacian spectra of its $k$-token signed graph. Besides, we introduce and study the unbalance level of a signed graph, which is a new parameter that measures how far a signed graph is from being balanced. Moreover, we study the relation between the frustration index and the unbalance level of signed graphs and their token signed graphs.

**Plot unscrambling.** Finally, we include a movie synopsis unscrambling task. We obtain movie plot synopses from IMDb or Wikipedia for feature-length films released after January 1st 2024. These synopses are then split into their constituent sentences and are randomly shuffled. The lengths of the synopses vary from as few as 7 sentences to as many as 66 sentences; at the upper end, this is a very challenging task. The LLM is provided the shuffled sentences with the prompt: 'The following plot

Table 13: Summary for tasks in `LiveBench`.

| Category | Task | Num. | Data Source |
|---|---|---|---|
| data_analysis | cta | 50 | Kaggle and Socrata datasets |
| data_analysis | tablejoin | 50 | Kaggle and Socrata datasets |
| data_analysis | tablereformat | 50 | Kaggle and Socrata datasets |
| instruction_following | summarize | 50 | The Guardian articles |
| instruction_following | paraphrase | 50 | The Guardian articles |
| instruction_following | story_generation | 50 | The Guardian articles |
| instruction_following | simplify | 50 | The Guardian articles |
| language | typos | 50 | ArXiv abstracts |
| language | connections | 50 | NYT daily puzzles |
| language | plot_unscrambling | 40 | IMDb plot synopses |
| reasoning | web_of_lies_v2 | 50 | N/A (synthetic) |
| reasoning | zebra_puzzle | 50 | N/A (synthetic) |
| reasoning | spatial | 50 | N/A (manually created) |
| math | olympiad | 36 | International Olympiad 2024 |
| math | AMPS_Hard | 100 | N/A (synthetic) |
| math | math_comp | 96 | AMC 2023 and AIME 2024 |
| coding | coding_completion | 50 | LiveCodeBench |
| coding | LCB_generation | 78 | LiveCodeBench and Leetcode |

summary of a movie has had the sentences randomly reordered. Rewrite the plot summary with the sentences correctly ordered. Begin the plot summary with <PLOT_SUMMARY>.'.

Scoring the task involves two decision points: 1) how to deal with transcription errors - those in which the model modifies the lines when producing its output 2) given the ground truth ordering of sentences and the LLM's ordering, choosing an appropriate scoring metric. For 1), one option is to permit only strict matching – that is, the LLM must transcribe perfectly. However, although the strongest models do perform well on this (we find they achieve over 95% transcription accuracy), we find that LLMs often correct grammatical errors or spelling mistakes in the source data when transcribing. As we are primarily interested in testing the models' capabilities for *causal language reasoning* in this task, rather than precise transcription accuracy, we instead apply a fuzzy-match using difflib (Team, 2008) to determine the closest match using a version of the Ratcliff/Obershelp algorithm (Ratcliff & Metzener, 1988). For 2), we calculate the score as $1 - \frac{d}{\text{n\_sentences\_gt}}$, where n_sentences_gt is the number of sentences in the ground truth synopsis, and $d$ is the Levenshtein distance (Levenshtein, 1966) of the model's sentence ordering to the ground truth synopsis ordering. Thus if the model's sentence ordering perfectly matches the ground truth, the distance $d$ would be 0, and the score would be 1 for that sample.

One might think that it is plausible that synopsis unscrambling cannot always be solved with the information provided. However, note that even if the set of sentences do not create a distinct causal ordering, the task is essentially asking the LLM to maximize the probability that a given arrangement of sentences is a real movie. In addition to causal reasoning, the LLM can use subtle cues to reason about what ordering creates the most compelling plot. Furthermore, even if there does exist an upper bound on the score that can be achieved that is strictly below 100%, it can still be a useful metric for distinguishing models' relative strengths. An analogous metric is that of next-token perplexity in language modelling; although it is likely that a perfect prediction of the next token is impossible to achieve, and we do not even know what the obtainable lower bound on perplexity is, it is still a powerful metric for determining language-modelling performance.

Example questions from the plot unscrambling task can be lengthy, so examples can be viewed in our repo here.

In Table 13, we also present a summary table consisting of the number of questions and source data for each task in `LiveBench`.

### A.4    GRADING METHODOLOGY

LiveBench makes use of automatic regex-based grading methods in order to avoid the biases and other downsides of LLM judging, as detailed in Section 1. On the other hand, automated grading methods have two main pitfalls, which we take care to circumvent. The first is that an instruction-following element is added to the task. For example, a reasoning task with a complex answer instruction format tests instruction-following as well as reasoning, rather than pure reasoning. The second is that care must be made to allow for different answer formats so as not to favor a particular LLM (even after giving exact, unambiguous instructions in the prompt, regarding the format of the answer).

When creating questions, we make sure that the prompt is written in such a way that the answer format is fully specified and unambiguous. We typically give an example of the answer format in the prompt, and specify any potential parts that are unclear.

Additionally, our grading methodology is designed to be fully permissive, accommodating all answer formats within reason. As a rule of thumb, if a human reading the LLM's answer would be able to comprehend the answer (and assuming it is correct), then the answer should be marked correct. For example, in our math category, models can respond with answers in either latex 'boxed{}' or 'fbox{} ' environments. For multiple choice questions, we ask for the letter answer, but we accept either the letter or the raw answer. The reason for our less-strict judging is because some models may have been instruction-tuned to a particular format, giving them an advantage or disadvantage. We achieved this by using flexible regex patterns that capture different notation styles or response structures. This approach allows us to evaluate models based on their task-specific skills, rather than their ability to follow specific instructions.

In order to ensure a given task is not overly testing instruction following (in addition to its actual category), and also to ensure scoring accuracy and fairness, we regularly manually inspect a random sample of responses which were labeled incorrect for each model. We look for common answer formats that are incorrect and not picked up by the regex-based parser. This allows us to regularly improve our automatic scoring functions to admit as many answer formats as possible, within reason.

We continue to monitor model outputs and update scoring functions as needed to capture answers appropriately. When adding new models or tasks to the benchmark, we re-evaluate and refine scoring methodologies to ensure fairness and accuracy. This ongoing evaluation and maintenance process enables us to maintain the integrity of our benchmark, providing a comprehensive assessment of models' strengths and weaknesses. By decoupling instruction following capabilities from task-specific evaluations, we can accurately evaluate and compare models across various tasks and categories.

### A.5    QUALITY CONTROL FOR NEW MODELS

When integrating new models into LiveBench, we conduct thorough quality control to ensure consistent and accurate evaluation. This process guarantees that our benchmark remains reliable and effective in assessing model capabilities.

**New Model Setup**    We perform all setup in order to accurately run inference on the model. New models come with a README or example code on the chat template, system message, and hyperparameters to run the model, and so we match this setup in LiveBench's code. For API models from an existing model family, there is typically very little additional setup. For an API model from a new family, such as the O1 series, we add the same hyperparameters, system message, etc., as its example code. Similarly, for open-weight models, we make sure to match the published example code. Note that we generally prefer to use (trusted) APIs, even for open-weight models, because APIs are more controlled and fewer things can go wrong. As an example, new models occasionally have bugs when they are first released.

**Model Output Evaluation**    Next, we evaluate the new model on LiveBench and compare the new model's scores to other models of similar average score, or models from the same family or base. We look for tasks that scored unusually low, relative to similar models. This is not a comprehensive test, but it helps to more quickly flag tasks that have an unexpectedly low result, to debug potential errors in the parsing code.

**Manual Verification**    Next, we manually inspect the incorrect answers from the flagged tasks above, as well as a random sample of the responses across all tasks. The goal is to make sure that the wrong answers are truly incorrect, rather than a correct answer in the wrong format, or in such a way that it is marked incorrect by the parsing function. (Note that these checks are focused on catching false negatives. It is extremely unlikely for a false positive to occur, but we checked for false positives in the initial development of LiveBench, and occasionally check answers that are scored correct, for this reason.)

**Scoring Function Updates**    Based on our evaluation and verification, we update our scoring functions to accommodate any unique response patterns or notation styles exhibited by the new model. This maintains the permissiveness and fairness of our grading methodology, ensuring accurate scoring. Note that we made a few such updates when new models came out that hit new edge cases of our scoring functions, in the first month of LiveBench's release, but now we have not needed to make a scoring function update for a few months.

**Re-Running the Evaluation Pipeline**    After potentially updating scoring functions for some tasks, we re-score all models with the new scoring functions. This ensures consistent and accurate results.

### A.6    QUESTION UPDATE POLICY

To ensure the continued relevance and effectiveness of LiveBench, we have implemented a thorough question update policy. That said, we still maintain a degree of flexibility in updating questions, as well as updating the LiveBench policies as a whole, in order to be able to adapt to new developments and trends in the field (such as the release of models that employ search at inference time, or the popularity of a new type of LLM capability). In this section, we give more details on the update policy laid out in Section 2.7.

**Regular Updates**    We update questions and tasks on a monthly basis, incorporating new and challenging prompts that reflect the evolving landscape of large language models.

In each update, we replace $1/6$ of the questions on average, so that the benchmark is fully refreshed roughly every 6 months. We may speed up the turnover rate of questions in the future, based on interest in `LiveBench`. Each month, we do not release the new questions until one month later, so that the public leaderboard always has $1/6$ questions that are private and completely contamination-free.

We choose tasks to update based primarily on two factors: *(1)* the oldest tasks, and *(2)* the currently easiest tasks. In this way, the questions in `LiveBench` will stay new and continue to challenge the most capable LLMs. We also update tasks if there is any suspected contamination or other reason for updating a task. Nearly all tasks can be updated simply by running a script. See Table 14 for a breakdown of each task. Although most tasks can be auotmatically updated with a script, we always modify the generation script when creating new questions, in order to make sure that the distribution of questions is different and harder over time.

**Evaluation Re-Runs**    After updating questions and tasks, we re-run the evaluation pipeline on all models, ensuring consistent and accurate results. By implementing this comprehensive question update policy, we guarantee that LiveBench remains a vibrant, dynamic, and relevant benchmark for evaluating large language models.

**Version Control**    We maintain version control over all updates, ensuring transparency and reproducibility. This allows researchers to track changes and compare results across different versions of LiveBench.

**Community Engagement**    We encourage community engagement and collaboration in expanding and improving LiveBench. Researchers and developers can contribute new tasks, provide feedback on existing ones, and participate in shaping the benchmark's evolution.

Table 14: Description for whether tasks in `LiveBench` can be updated automatically.

| Category | Task | Can be updated automatically using a script? |
|---|---|---|
| data_analysis | cta | Yes - replace with newer datasets |
| data_analysis | tablejoin | Yes - replace with newer datasets |
| data_analysis | tablereformat | Yes - replace with newer datasets |
| instruction_following | summarize | Yes - replace with newer articles and/or update synthetic generation script |
| instruction_following | paraphrase | Yes - swap out news articles and/or update synthetic generation script |
| instruction_following | story_generation | Yes - swap out news articles and/or update synthetic generation script |
| instruction_following | simplify | Yes - swap out news articles and/or update synthetic generation script |
| language | typos | Yes - replace with newer paper abstracts |
| language | connections | Yes - replace with newer (daily) puzzles |
| language | plot_unscrambling | Yes - replace with newer imdb plot summaries |
| reasoning | web_of_lies_v2 | Yes - rerun (and/or update) the synthetic generation script |
| reasoning | zebra_puzzle | Yes - rerun (and/or update) the synthetic generation script |
| reasoning | spatial | No - create new questions by hand |
| math | olympiad | Yes* - rerun (and/or update) synthetic generation script, but source can only be updated yearly |
| math | AMPS_Hard | Yes - rerun (and/or update) the synthetic generation script |
| math | math_comp | Yes* - but can only be updated yearly, or by using different contests |
| coding | coding_completion | Yes - replace with more recent LeetCode questions |
| coding | LCB_generation | Yes - replace with more recent LeetCode questions |

## A.7 CONTAMINATION IN LIVEBENCH

We point out two different definitions of contamination: (1) *test set contamination*, and (2) *task contamination* – or train test distribution similarity.

The first definition is the one that we use in Section 1: when the test data (the questions and answers from the benchmark) are present in the training data itself. This is the definition that is commonly used in the literature, often called 'data contamination', 'data leakage', 'test set contamination', or even simply 'contamination' (Oren et al., 2023; Singh et al., 2024; Kalal et al., 2024; Golchin & Surdeanu, 2023a).

The second definition is related to the question: how well do LLMs generalize today? A low level of generalization is, e.g., training on AMC questions, and generalizing the same questions where the order of the answer choices, and other prose in the questions, are changed, while a high level of generalization is, e.g., training on AMC 12A 2023 and testing on AMC 12A 2024. The latter example, despite LLMs already exhibiting some degree of this type of high generalization, is still 'fair game' with respect to pretraining and fine-tuning: many LLMs are trained on competitive math problem tasks and then evaluated on new, unseen test problems from the same distribution.

Despite it being accepted practice, we attempt to guard against excessive uses of (2): we do not publicly release any of the code used to generate LiveBench questions, we always modify the generating code when updating the LiveBench questions (making the questions harder), and we do not publicly release the new questions for one month.

We also acknowledge that LiveBench does *not* fully satisfy (1): while nearly all questions are from June 2024 or more recent, there are some coding questions from November 2023, and the AMC questions have only undergone a low level of modification from their November 2023 version. Therefore, a limited fraction of LiveBench is likely contaminated on all recent LLMs.

## B ADDITIONAL DOCUMENTATION

In this section, we give additional documentation for our benchmark. For the full details, see https://github.com/LiveBench/LiveBench/blob/main/README.md.

## B.1 AUTHOR RESPONSIBILITY AND LICENSE

We, the authors, bear all responsibility in case of violation of rights. The license of our repository is the **Apache License 2.0**.

Table 15: Statistics for tasks in `LiveBench`. This table gives the number of questions for each task, as well as the mean and std. dev number of output tokens per question for `gpt-4-turbo-2024-04-09`.

| Category | Task | Num. | Input Tokens | | Output Tokens | |
|---|---|---|---|---|---|---|
| | | | Mean | Std. Dev. | Mean | Std. Dev. |
| data_analysis | cta | 50 | 879.52 | 1294.81 | 2.42 | 1.36 |
| data_analysis | tablejoin | 50 | 3111.28 | 1186.58 | 58.06 | 35.03 |
| data_analysis | tablereformat | 50 | 1395.54 | 814.92 | 475.64 | 333.57 |
| instruction_following | summarize | 50 | 1818.92 | 462.22 | 218.46 | 110.38 |
| instruction_following | paraphrase | 50 | 1805.62 | 501.30 | 267.98 | 101.98 |
| instruction_following | story_generation | 50 | 1794.30 | 474.77 | 405.24 | 156.49 |
| instruction_following | simplify | 50 | 1835.42 | 486.83 | 226.00 | 114.61 |
| language | typos | 50 | 1291.72 | 402.04 | 207.24 | 74.25 |
| language | connections | 50 | 884.54 | 48.94 | 422.06 | 541.96 |
| language | plot_unscrambling | 40 | 3545.68 | 1333.37 | 613.70 | 166.17 |
| reasoning | web_of_lies_v2 | 50 | 1670.64 | 337.40 | 490.82 | 98.68 |
| reasoning | zebra_puzzle | 50 | 1407.20 | 281.02 | 616.94 | 105.61 |
| reasoning | spatial | 50 | 450.48 | 134.30 | 337.88 | 88.16 |
| math | olympiad | 36 | 4646.08 | 1579.41 | 1221.22 | 685.41 |
| math | AMPS_Hard | 100 | 193.86 | 91.39 | 561.26 | 249.80 |
| math | math_comp | 96 | 678.20 | 243.09 | 664.27 | 424.33 |
| coding | coding_completion | 50 | 2659.30 | 718.70 | 67.08 | 46.00 |
| coding | LCB_generation | 78 | 2159.88 | 796.14 | 213.44 | 154.34 |
| Total | | 1000 | 1612.27 | 703.73 | 394.85 | 261.79 |

## B.2 MAINTENANCE PLAN

The benchmark is available on HuggingFace at https://huggingface.co/livebench.

We actively maintain and update the benchmark, already having added multiple updates, and we continue to welcome contributions from the community.

## B.3 CODE OF CONDUCT

Our Code of Conduct is from the Contributor Covenant, version 2.0. See https://www.contributor-covenant.org/version/2/0/code_of_conduct.html.

## B.4 DATASHEET

We include a datasheet (Gebru et al., 2021) for LiveBench in https://github.com/LiveBench/LiveBench/blob/main/docs/DATASHEET.md.

## B.5 BENCHMARK STATISTICS

Here, we give statistics on the number of questions and average number of output tokens per task, and the total cost of running `LiveBench` with common API models. For the number of questions for each task, as well as the mean and std. dev number of input and output tokens per question for `gpt-4-turbo-2024-04-09`, see Table 15. Across the 1000 questions in our current problem set, the mean number of input tokens per question was 1612, and the mean number of output tokens was 395. See Table 16 for the price to run GPT and Claude models on `LiveBench`, as of October 1, 2024. `o1-preview-2024-09-12` is the most expensive at $47.87, while `claude-3-haiku` is the cheapest, at $0.90.

Table 16: Prices for running GPT and Claude models on `LiveBench`. This table gives the approximate cost for running models on `LiveBench` as of Oct 1, 2024. Note that we used the `gpt-4-turbo` tokenizer for all computations, so all other prices are approximate.

| Model | Price in USD |
|---|---|
| o1-preview-2024-09-12 | $\approx 47.87$ |
| o1-mini-2024-09-12 | $\approx 9.57$ |
| gpt-4o-2024-05-13 | $\approx 13.98$ |
| gpt-4-turbo-2024-04-09 | 27.97 |
| gpt-4-1106-preview | $\approx 27.97$ |
| gpt-3.5-turbo-0125 | $\approx 1.40$ |
| claude-3-opus | $\approx 53.80$ |
| claude-3-5-sonnet | $\approx 10.76$ |
| claude-3-sonnet | $\approx 10.76$ |
| claude-3-haiku | $\approx 0.90$ |

