# OpenReview forum: "LiveBench: A Challenging, Contamination-Limited LLM Benchmark"
_ICLR.cc/2025/Conference — ICLR 2025 Spotlight_

### Official Review · Reviewer_G9nN · 2024-10-30

**Soundness:** 3
**Presentation:** 3
**Contribution:** 4
**Rating:** 6
**Confidence:** 3

**Summary:**

The paper presents LiveBench, a new benchmark for evaluating LLMs designed to avoid common issues such as test set contamination and biases from LLM or human judging. LiveBench achieves this by (1) frequently updating questions with information from recent sources, (2) employing automated scoring against objective ground-truth values, and (3) covering a wide range of challenging tasks in areas such as math, coding, reasoning, language, instruction following, and data analysis.

The authors evaluated multiple prominent closed-source and open-source LLMs, showing that even top-performing models struggle to achieve over 70% accuracy, highlighting the benchmark’s difficulty.

**Strengths:**

1. This paper provides a dynamic, comprehensive and challenging benchmark that stays current with LLM development.
2. The benchmark’s monthly updates and open access to questions, code, and model answers contribute to high reproducibility and transparency.

**Weaknesses:**

1. The paper fails to analyze the quality of the questions used for evaluation, particularly those manually constructed or adjusted for model testing.
2. The paper seems to employ a regular expression matching approach against ground truth to evaluate model outputs, a method which can introduce evaluation bias, especially for models with weaker instruction-following abilities. While the paper mentions potential biases from using large language models as evaluators, it neglects to analyze how regex-based matching could exacerbate evaluation inaccuracies.

**Questions:**

1. Are all tasks' ground truths unique and definitive? How do you ensure that there is no ambiguity in the ground truth for each task?
2. In line 509, you mentioned that "In Srivastava et al. (2024), the authors modify the original MATH dataset (Hendrycks et al., 2021) by changing numbers in the problem setup, leading to significant declines in model performance across all LLMs, including the leading ones." Given this, do the tasks generated in this benchmark, such as the Typos task, also face similar issues due to using updated data sources under the same modification rules?
3. Could you provide a statistical table that clearly presents the number of questions, tasks, sources, and other relevant information for each category? This would enhance clarity and help readers understand the distribution and characteristics of the tasks more effectively.

---

> ### Author Response · Authors · 2024-11-25
>
> Thank you for your detailed review. We appreciate that you found our work has high reproducibility and transparency, and that its contributions are excellent. Further, we appreciate that you found our benchmark comprehensive, challenging, and dynamic. We address each of your questions below:
>
> **W1: on analyzing the quality of the questions.**
> Thank you for the suggestion! We do ensure that all of the questions and evaluations in LiveBench are very high quality. We have now added sections A.4: Grading Methodology and A.5: Quality Control for New Models to our appendix, which details all of the steps we take to ensure that our questions are high quality. There have been several times when the evaluation of a new model causes us to find and correct an additional edge case in our scoring functions. We also develop the questions with quality control in mind. For example, for the spatial reasoning task (the only task that was not synthetically generated, so each question must be checked individually), we gave the questions to three humans to forerun, who caught and corrected multiple small mistakes and ambiguities.
>
> LiveBench’s design of regularly updating the questions also gives us the unique opportunity to regularly perform more quality control and further improve the questions. For example, based on the feedback from Reviewer uo5E, we changed the answer format for connections and zebra puzzles to be XML tags, so that the parsing script will not confuse the answer with other markdown formatting.
>
> **W2: On Regex matching.**
> This is a great point! We definitely recognize the interplay between using regex-based grading methods and requiring a model to be sufficiently adept at instruction following. Since we do have a specific instruction following category, for regex-grading on other categories we strive to make the grading be permissive to various forms of providing the answer. For example, for math questions, the prompt asks for answers to be in a “\boxed{}” environment. However, we saw that some models responded with multiple other formats, so we added these as possibilities to be scored correctly.
>
> In general, our process for validating the regex matching to minimize scoring biases and not force instruction following is to:
> - Make a grading methodology which aligns with the prompt and task
> - Generate the model outputs and scores
> - Manually inspect a random sample of responses and scores for each model
> - Update the scoring method when needed, to be permissive and fair to all models
> - Additionally, when we add new models to the benchmark, we will also look at their generations and ensure that the scoring functions are capturing answers appropriately and update the scoring function if necessary.
>
> So this methodology is our standard procedure to balance a permissive, yet fair, scoring function with the various instruction following capabilities of different models.
>
> **Q1: Are all ground truths unique and definitive?**
> This is a great question. Most of the questions have unique and definitive ground truths, but all of the scoring is task-specific, and there are some exceptions. For the AMPS_Hard task (in the math category), we accept all answers that are mathematically equivalent to the ground truth answer, using the sympy package (our code here is inspired by the [Eleuther AI eval harness](https://github.com/EleutherAI/lm-evaluation-harness), the same code used to score GSM8K and MATH on the Huggingface leaderboard). For the instruction-following tasks, we do not match with the ground truth; instead, we run code to check whether the answer adheres to the instructions, such as formatting the output or including keywords.
>
> **Q2: On modifying questions.**
> In Srivastava et al. (2024), the authors modified the original MATH dataset in order to find evidence that some models were over-tuned to the original MATH dataset and unable to generalize well. For the typos task, we are taking a very recent arxiv abstract, and introducing typos, and asking the LLM to fix the typos. As with all LiveBench questions, we will regularly update the source data (the arxiv abstracts). So, it is a different situation compared to Srivastava et al. (2024).
>
> **Q3: New statistical table.**
> Yes, thank you for the suggestion! We have now created Table 13 that presents the category, task, number of questions, and source data for each task. (Note also that Table 14 and Table 15 provide additional information as well.)
>
>
> Thank you very much, once again, for your excellent comments. We respectfully ask that if you feel more positively about our paper, to please consider updating your score. If not, please let us know what can be further improved; we are happy to continue the discussion any time until the end of the discussion period.

---

> > ### Comment · Reviewer_G9nN · 2024-11-26
> > **Official Comment by Reviewer G9nN**
> >
> > Thank you for addressing my concerns with detailed responses. Most of my concerns have been resolved, and thus I will maintain my positive score.

---

> > > ### Author Response · Authors · 2024-11-28
> > >
> > > Thank you! We are glad that we addressed most of your concerns. Please let us know if you have any more questions or suggestions before the end of the rebuttal period, and we will be happy to respond to them.

---

### Official Review · Reviewer_L2M4 · 2024-11-03

**Soundness:** 4
**Presentation:** 4
**Contribution:** 4
**Rating:** 8
**Confidence:** 3

**Summary:**

The paper presents LiveBench, a benchmark designed to evaluate LLMs while avoiding data contamination and biases associated with LLM and human judges. LiveBench includes frequently updated, diverse tasks across categories such as math, coding, reasoning, and data analysis, scoring responses based on ground-truth answers to ensure objectivity. With new, harder tasks released monthly, LiveBench aims to accurately assess LLMs as they improve over time.

**Strengths:**

- The benchmark addresses a prevalent issue in LLM evaluation.
- The paper is well-written, clearly explaining both the motivation and the implementation.
-  LiveBench scores models against objective ground-truth answers, not relying on possibly-biased LLM-judges.
- Tasks presented are very diverse.

**Weaknesses:**

- Some models' performance in evaluations are very close. It's hard to say how similar they are without a significance test / stds.
- As LLM developers may train their models to excel on specific benchmarks, frequent updates could eventually lead to overfitting to LiveBench’s task formats, even with the regular introduction of new content. This could incentivize models to optimize for LiveBench-specific tasks rather than improving general capabilities. This can be solved with the introduction of more types of tasks and the contribution of the community, but it's also worth thinking about refreshing existing tasks' formats.

**Questions:**

Why did you not evaluate with some randomness to obtain significance over noise?

---

> ### Author Response · Authors · 2024-11-25
>
> Thank you for your detailed review. We appreciate that you found our work addresses a prevalent issue, that the tasks are very diverse, and that our benchmark does not rely on possibly biased LLM-judges. Further, we appreciate that you found our work well-written, with clear explanation for the motivation and implementation. We address each of your comments below:
>
> **W1: Models’ performance are close.**
> Thank you, that is a great point! Note that Figure 1 does include 95% bootstrap confidence intervals, and can see that e.g. assuming 95% confidence, o1-preview achieves the best overall score, and then there is no statistically significant clear 2nd place model: the confidence intervals of the next four models are all overlapping (especially claude-3-5-sonnet-20240620 and o1-mini-2024-09-12). To make this clearer, we will add a table that includes the confidence intervals in the table.
>
> **W2: On overfitting to LiveBench task formats.**
> Thank you for this point, this leads to a great discussion! We agree that there is risk of overfitting to LiveBench’s task formats over time. Based on your suggestion, for our latest update, we did indeed update the format of some existing tasks, such as the connections task and the zebra puzzle task. The good thing about LiveBench is that our update policy is flexible and designed to adapt with the field, and to overcome new issues that come up over time. For example, for our updates so far, some tasks have been particularly novel (such as the new spatial reasoning task), while other tasks have been smaller changes to existing tasks (such as the updated math competition task).
>
> **Q1: on obtaining statistical significance.**
> Answered in W1.
>
> Thank you again for your excellent comments. We are happy to continue the discussion any time until the end of the discussion period. Thank you!

---

> > ### Comment · Reviewer_L2M4 · 2024-12-01
> >
> > Thank you for your answer, you've addressed my concerns. Good luck!

---

### Official Review · Reviewer_uo5E · 2024-11-04

**Soundness:** 2
**Presentation:** 3
**Contribution:** 3
**Rating:** 8
**Confidence:** 4

**Summary:**

The paper introduces LiveBench, a new benchmark designed to evaluate LLMs in a way that mitigates common issues like test set contamination and biases introduced by human or LLM judges. LiveBench is continuously updated with new questions sourced from recent information, such as math competitions, arXiv papers, news articles, and datasets. This ensures the benchmark remains relevant and resistant to contamination by models trained on static datasets. Unlike many benchmarks that rely on subjective human or LLM judgments, LiveBench uses objective ground-truth values for scoring. LiveBench includes 18 tasks across six categories—math, coding, reasoning, language comprehension, instruction following, and data analysis. Each task is designed to be difficult, with top-performing models achieving less than 70% accuracy. The tasks are either based on recent real-world data or are harder versions of existing benchmark tasks. The paper evaluates dozens of LLMs, both proprietary and open-source. The benchmark provides detailed results across all six categories and highlights the strengths and weaknesses of each model. The authors plan to expand the benchmark with new tasks and models over time. They also plan regular updates to ensure the benchmark remains challenging as LLMs improve.

**Strengths:**

Originality:
This paper makes very well rounded verifiable benchmarks that cover a large number of tasks in different ways. This in itself is unique and a great contribution to LLM research.
While some of the tasks are inspired by existing benchmarks, some seem to be completely novel evaluation strategies (such as the proof reconstruction).

Quality:
The benchmark itself seems mostly well constructed and fair. The emphasis on task diversity and verifiability is commendable.

Clarity:
The methods are generally clear. Experiments are well explained. Benchmark tasks/evaluation are clear.

Significance:
This paper gives a useful benchmark for offline evaluation of LLMs. It has more task coverage than many other offline evaluation schemes. It also presented potential solutions to the benchmark contamination problem by making the benchmark “live”.

**Weaknesses:**

The core feature LiveBench claims to offer is being “contamination-free” However, it seems this claim rests upon two main assumptions that are never addressed:
1. The source data is not contaminated.
2. The data perturbations used to decontaminate or generate new data are actually sufficient to do so.

For (1): Consider the math category which utilizes questions from AMC12 2023, SMC 2023, AIME 2024, and IMO 2024. It seems likely these high quality data points would already be exploited by model developers for training models, intentionally or unintentionally. While the author’s do not directly discuss the implications of this, they do mention they do make some modifications to the questions. This leads to (2).

For (2): “Contamination-free” is a strong statement. **First and foremost, I recommend the authors define exactly what they mean by “Contamination-free”.** As a reader, my understanding of “contamination-free” is as follows: *Given I train my LLM on LiveBench up to month M, I will have no performance gain on the queries released in month M + 1.* Now, if the authors choose to run with this relatively strong definition of contamination-free, the authors should consider experimentally verify that training a model on prompts up to month M leads to negligible gains on prompts from month M + 1. Right now, I find this doubtful. For example, one of the author’s stated methods to change queries is through “modifying the prose and the answer order” (line 142). While it is likely that changing trivial sections of a query may decrease an overfit LLM’s performance slightly, the authors do not show that this fully mitigates contamination, or even to what extent it does (if not fully). Moreover, the authors suggest on line 303 that “many of the tasks are synthetic and therefore very fast and simple to create a new set of questions based on fresh data”. This raises the question: what is stopping an adversarial from generating, let’s say 1 million rows, of LiveBench data through this “fast and simple” process, training on it directly. Does this count as “contamination”? Would the release of 200 prompts in month M + 1 protect against this? Alternatively, let’s consider the instruction following summarization task: in this case, the update process presumably is to swap out the Guardian articles with new ones and change around the instructions selected from the 16 choices. However, how can the author’s be sure this meaningfully changes the task, especially because the article doesn’t even matter— the only thing the model needs to learn in order to overfit is the 16 different instructions that it will be checked on?

The authors could potentially address this in 2 ways:
1. Experimentally show how much training (with a reasonable setup) on M affects M + 1.
2. Establish a very weak definition of “Contamination-free” (e.g. the exact prompt is not seen in M + 1). However, there are no guarantees that training on M will not help M + 1. (Note that this heavily decreases the significance of the benchmarks as “contamination-free”)

**Bottom Line: In its current implementation, this paper has little evidence that its current query perturbation/generation process has any effectiveness against contamination. The authors should first define contamination, then defend that their benchmark sufficiently protects against this definition of contamination.**


**The “live” aspect of the benchmark is powered by brute force, and is not necessarily automatic in any sense:** It is not entirely clear how LiveBench is significantly more “update-able” than existing static and verifiable benchmarks, such as MMLU or IFEval. The author’s pledge to keep updating the benchmark monthly does not seem like a sufficient novel contribution to benchmark construction in general. The authors should document the update plan for every single fine-grain task. They should explain the estimated time and monetary cost to update this specific task. They should detail any optimizations in task design to ensure it is easy to update, if any. This would be a great table for the appendix and would make it clear what the paper contribution on this front is.


**Ablations on LLM-Judgments are insufficient, and potentially misleading:**
The experiments to show “LLM judges cannot accurately evaluate challenging math and reasoning questions” are not enough to address the scope of the claim.
In particular:
1. Only GPT-4-Turbo was tested as judge
2. GPT-4-Turbo was mainly judging itself, which is already likely worst case scenario
3. Only 2 models total were judged
4. Only 4 benchmarks tested
5. Subpar judge prompt

    a. No reference answer (generally these judges have either pairwise comparisons, or some form of reference answer)

    b. No rubric for judging

Possible ways to strengthen the experiment to adequately support the claim are as follows:
1. Utilize more LLMs as judges
2. Use the LLM to judge more models
3. Judge with a reference answer, and/or use a more updated judge prompt, such as the judge prompt from Arena-Hard-Auto or Alpaca-Eval 2.0.
4. Show accuracies on all tasks


**Insufficient comparisons to other benchmarks or “ground truth” evaluations:**
The authors compare LiveBench to Chatbot Arena and Arena-Hard. They hypothesize that the differences in rankings are mainly due to style or self-bias. If the authors suspect style to be a large factor in ranking difference they should compare LiveBench taking to style controlled Chatbot Arena rankings, proposed in the blog https://blog.lmarena.ai/blog/2024/style-control/. Note that Arena-Hard also has style control implemented. The authors should provide a table showing the correlation between the benchmarks. There are also Chatbot Arena categories, such as Math or Instruction following, which would provide more informative correlations to the respective LiveBench tasks.

**Questions:**

On line 214, the author suggests that LLMs are commonly used to perform table joins. Can they elaborate on this? This does not seem like a common real-world use case.


On line 299, the authors suggest they will release 200 prompts per month, but it seems in lines 307-311 they released significantly less than 200 prompts per month. Could the authors clarify what the plan for questions releases is?


On line 331, the authors suggest they use FastChat chat templates. Why is FastChat used over the explicit template provided in model tokenizers or readmes? FastChat is not necessarily guaranteed to be maintained to be constantly up to date for this purpose. Not all models are implemented in FastChat.


On line 363-365, the authors suggest that the zebra puzzles are too hard such that top models do not perform better than random chance. If this is the case, is there anything that can be done to make this category more relevant? E.g, making it slightly easier? Adding this noise into a benchmark seems unnecessary.


On line 402-407, the authors color outliers in red. However, outlines that are far below the line (large negative residual) are not colored. Why were some points determined as outliers but others not? This is true for both graphs.  Moreover, on the left plot, o1-preview is shown as an outlier, but the point next to Phi-3.5-mini-instruct is not shown as an outlier, even though the latter is even farther from the line of best fit.


On lines 438-445, the authors show Starling-LM-Beta on their graph, presumably to show the large differential between Chatbot Arena score and LiveBench score. However, Starling-LM-Beta is notably missing from Figure 5 which is described to show “all models”.  It is also missing from Table 3. Starling appears again in Figure 6… then it is gone again in Table 7… then back again in Table 9. Why is this the case? (I think Open Hermes and a couple other models also appear and disappear).


On lines 324-329, the authors cite various models they use. However, the citations for the remaining models are not present anywhere else. I find it strange that some models are chosen to be cited properly and others are not.


It seems the authors commonly use **answer ** to make the LLM generated answer parsable for verification. Why was ** chosen here, when it is also a common markdown symbol? Wouldn’t a less common symbol make sense here, like “{{answer}}” as one example. Peaking at the code, which seems to take the last bolded text as the final answer, it is conceivable that a markdown heavy model could have some extra bolding after the answer and get penalized for this. Think of something like “... **Final Answer: ** **X **\n\n **Additional Considerations: ** …” It is important to note that adding any bolded content after the final answer seems to not be explicitly warned against in the prompt to the LLM.


On lines 914-917, the authors show the judge prompt used for the LLM judge, which outputs a numerical score 1-5. However, it is unclear how this score is turned into a binary label used to calculate the error rates seen in Table 2.

---

> ### Author Response · Authors · 2024-11-25
> **Reply to uo5E, Part 1**
>
> Thank you **very much** for your detailed review. First, we appreciate that you found our work to be unique, well-constructed, fair, and a great contribution to LLM research. Further, we are glad that you find the emphasis on task diversity and verifiablility to be commendable. For your other points, we found ourselves agreeing with all of your points and adding, or planning to add, nearly all of your suggestions. In particular, we have now added more discussion and formalism when talking about contamination and when making claims about the properties of LiveBench. We believe that this has made the paper significantly clearer and more sound. Thank you! Now we address all of your points individually:
>
> **W1: On the definition of “contamination-free”**
> We agree with all of your points about adding a formal definition of “contamination-free”, and then making a formal claim on whether LiveBench fits this definition. In summary, we are *not* able to claim that LiveBench is contamination-free, and we have updated the paper and the title accordingly. That said, as we will discuss in more detail later, we strongly believe that even with (limited) contamination, LiveBench is and will continue to be a significant contribution to the field.  In particular, its overall design decisions lead to much less contamination than most other reasoning-heavy benchmarks, while still having usability and transparency.
>
> We point out two different definitions of contamination: (1) test set contamination, and (2) *task* contamination -- or train test distribution similarity.
>
> The first definition is the one that we defined in lines 11-12 in our paper: when the test data (the questions and answers from the benchmark) are present in the training data itself. This is the definition that is commonly used in the literature, often called “data contamination”, “data leakage”, “test set contamination”, or even simply “contamination” [1-6].
>
> The second definition, related to the one that you refer to in your review, is interesting and related to the question: how well do LLMs generalize today? A low level of generalization is e.g. training on AMC questions, and generalizing the same questions where the order of the answer choices, and other prose in the questions, are changed (as we did for the AMC questions), while a high level of generalization is e.g. training on AMC 2023 and testing on AMC 2024. We note that the latter example, despite LLMs already exhibiting some degree of this type of high generalization, is still fair game with respect to pretraining and fine-tuning, e.g. all LLMs such as GPT-4 and O1 are trained on competitive math problem tasks and then evaluated on new, unseen test problems from the same distribution, which is viewed as a fair and useful metric [7]. (In the medium term, as LLM research advances and LLMs get better at generalization, it will become harder and harder to create useful benchmarks.)
>
> Synthetically generating data that is similar to LiveBench questions but which is not present in the LiveBench test set itself largely falls under (2) rather than (1), although the degree of (2) is often nuanced and depends on the task. We still guard against excessive uses of (2): we do not publicly release any of the code used to generate LiveBench questions, we always modify the generating code when updating the LiveBench questions (making the questions harder), and we do not publicly release the new questions for one month. This is still not a complete safeguard to (2), yet it makes it harder to gain an advantage in this way.
>
> All of that being said, we do acknowledge that LiveBench does *not* fully satisfy (1): while nearly all questions are from June 2024 or more recent, there are some coding questions from November 2023, and as described above, the AMC questions have only undergone a low level of modification from their November 2023 version. Therefore, a limited fraction of LiveBench is likely contaminated on all recent LLMs. We have updated our paper (including the title) accordingly, along with a discussion on the second type of contamination in Appendix A.6. As a final comment, we strongly believe that even with this (limited) contamination, LiveBench will be a significant contribution to the field. Designing a benchmark involves many design choices and trade-offs, both inherent and practical. For example, there is the decision on whether to use AMC math questions, with some modification, that are very high quality questions but are contaminated.

---

> ### Author Response · Authors · 2024-11-25
> **Reply to uo5E, Part 2**
>
> Static benchmarks that focus on reasoning or reasoning-related tasks, such as MMLU, GSM8K, or BigBench-Hard, are very popular within the community even today, despite being fully contaminated on LLMs for multiple years. LiveBench offers a reasoning-heavy benchmark with objective scoring, that has minimal contamination, especially when compared to static benchmarks. LiveBench gives the community a consistent look at how LLMs compare on reasoning and other types of tasks. There are other benchmarks with limited or no contamination, but they are substantially different and seek a different goal compared to LiveBench -- Chatbot Arena is excellent for what it does, but its focus is on crowdsourced, human-judged questions. The SEAL Leaderboards are an excellent resource for the community, but the questions are completely hidden, and the human-grading aspect means that they are unable to add more than about 20 models on the leaderboard, and they are unable to evaluate models anywhere close to the same speed as LiveBench. Therefore, LiveBench occupies an important area of contamination-limited, reasoning-focused leaderboard, and as a bonus, new models can be added very quickly to the leaderboard.
>
> [1] https://arxiv.org/abs/2310.17623
> [2] https://arxiv.org/abs/2411.03923
> [3] https://arxiv.org/abs/2311.06233
> [4] https://arxiv.org/abs/2411.02284
> [5] https://arxiv.org/abs/2311.09783
> [6] https://arxiv.org/abs/2405.00332
> [7] https://arxiv.org/abs/2303.08774
>
>
> **W2: The “live” aspect of the benchmark.**
> Thank you; we agree with your suggestions, and we have added these new details to the paper in Section A.6 and Table 13. For all but three tasks, we have private scripts that can create new questions immediately. Many use a web scraper of some sort, and all of them are quick to run (and have essentially no cost to run). Of course, re-running the LLMs on new questions does incur a cost. Using Table 15, we compute that fully rerunning all OpenAI/Anthropic models currently on the leaderboard (more than half of all costs) is `$255.38`. For an update that replaces one sixth of the questions, it costs `$42.50`. The institutions of the authors that maintain LiveBench can easily fund the labor and LLM costs for updating the leaderboard.
>
> As a “baseline effort”, the labor for the LiveBench updates could be done very quickly, just by running a few scripts. However, as described in our answer to W1, we always modify the generation scripts to decrease the similarity of the new questions to the previously released questions, and to make the questions harder over time. We also occasionally add brand new tasks, which take time to create. Furthermore, although again “baseline effort” updates would be easy to plan out far into the future, we intentionally do not fully plan out future updates in order to have the flexibility to adapt to new developments and trends in the field. For example, the O1 models were suddenly released and scored extremely well in three tasks: connections, zebra puzzles, and web of lies, making it clear that these tasks are now too easy for the best existing LLMs; we have now updated the first two to become harder, and we will update web of lies in the next release.
>
> LiveBench does not have “technical novelty”, but a better metric (both in the reviewer guidelines, and we believe, more generally) is impact and value to the community. For a benchmark to be successful and useful, it should be executed well, with high quality, and incorporate good design decisions. We believe that the core tradeoffs of LiveBench, such as the frequently updated questions, while still being open-source, and the focus on difficulty and quality of the questions, make LiveBench useful to the community as a benchmark.

---

> ### Author Response · Authors · 2024-11-25
> **Reply to uo5E, Part 3**
>
> **W3: Ablations on LLM-Judgments.**
> We agree with you, and in the paper, we have now clarified our statements in the paper and removed the overly broad statement that “LLM judges cannot accurately evaluate challenging math and reasoning questions.”
>
> Your discussion brings up interesting questions about the capabilities of LLM judges. The overall point that we attempted to make in the paper is obvious in retrospect (at least for a large class of problems): for many types of hard math and reasoning questions, if an LLM struggles to answer the question, then it will also struggle to determine whether or not a given answer to that question is correct. This statement is so intuitive, that in hindsight, it might have been clearer if we had given no experiments at all. Similarly, we believe that if an LLM judge is given access to the ground truth, then it is also obvious that the LLM would be able to judge the correctness of answers. (Note that in a setting with crowdsourced questions, there may not necessarily be access to ground truth anwers.) There is a large class of exceptions to the above statement: any problems that are hard to solve by frontier LLMs, yet easy to check whether an answer is correct or not. Most NP-Hard problems fall into this class.
>
> That said, all of the tasks in our original experiments (AMC, AIME, and Zebra puzzles) are in a gray area, where it is not clear whether solutions are easy or hard to check: for AMC/AIME questions, it is highly dependent on the specific question, and for Zebra puzzles, a correct answer reveals a small part of the solution.
>
> Regarding proposed experiments: we would be happy to run more experiments, although as mentioned, we now believe in hindsight that our claims are intuitive and may not need excessive experimental evidence.
>
> **W4: Additional comparisons to other benchmarks.**
> Thank you for the suggestion. We agree that this would make a great addition, and we are still working on completing these additional comparisons.
>
> **Q1: On table-joins.**
> Data analytics is a commonly named use cases for LLMs in industry (e.g. [1,2], which include several examples). Schema matching, entity matching, and CTA (another task in LiveBench), are all important data cleaning operations which enable valid joins / unions over noisy or mismatched data) [3-8].
>
> [1] https://blog.sumble.com/the-overlooked-genai-use-case/
> [2] https://www.youtube.com/watch?v=riWB5nPNZEM&t=1764s
> [3] https://openreview.net/forum?id=RMZVUP7NgL
> [4] https://arxiv.org/abs/2403.01567
> [5] https://arxiv.org/abs/2403.05266
> [6] https://arxiv.org/abs/2407.11852
> [7] https://arxiv.org/abs/2310.11244
> [8] https://arxiv.org/abs/2410.24105
>
> **Q2: Releases each month.**
> This is a good point! First, it is a slight typo: we target an average of one-sixth or 167 updated questions for each update. The first update consisted of adding 90 questions so that the number of total questions reached 1000. After that first update, we have focused on the tasks that are in most need of an update. So for example, the olympiad section, despite having among the highest signal of any task, has only 36 questions, so this decreased the total number of updated questions for that release. Our new release mentioned in the global comment consists of 300 updated questions.
>
> **Q3: Fastchat templates.**
> Yes, you are exactly right. Whenever we add a new model to LiveBench, we make sure that the chat template exactly matches the template provided in the readme of the new model. This has involved several instances of us manually updating the FastChat code that is within the LiveBench repo, for example, to add o1 [in line 264 here](https://anonymous.4open.science/r/LiveBench/livebench/common.py) or to add the phi models [in line 2419 here](https://anonymous.4open.science/r/LiveBench/livebench/model/model_adapter.py).
>
> FastChat is very useful as a framework for storing and applying the various chat templates of LLMs (and, at the time of LiveBench’s creation, FastChat was relatively up to date), but we make sure to update the code whenever we add a new model. We recently added the [conversation.py](https://anonymous.4open.science/r/LiveBench/livebench/conversation.py) file from fastchat directly into the LiveBench codebase (similarly to what we had already done with the model_adapter.py file). So, we use it for the purpose of having a framework and tooling to apply chat templates, and we ensure that every model uses the correct chat template.

---

> ### Author Response · Authors · 2024-11-25
> **Reply to uo5E, Part 4**
>
> **Q4: Zebra puzzles.**
> Yes, we said “all but the highest-performing models do not perform significantly better than random chance”: yet the highest-performing models do perform quite well (o1-mini performs the best at 82%), and most models perform better than random chance, just not “significantly better” (after the top four, the average of the 20 next-best models is 41%, while random chance is 33%). So we do get some signal from this dataset, especially for top models.  Still, there is a good deal of noise.
>
> We agree with you, and in our most recent update, we have now updated Zebra puzzles. We actually had to make it slightly harder, since a key tenet of LiveBench is that it is challenging for even the current best models which people are likely to be comparing, and o1-mini had achieved 86%. However, a key update was to modify the prompts such that we can give partial credit, which significantly reduces the noise.
>
> **Q5: On showing outliers systematically.**
> Thanks for pointing this out! We are working on updating all figures in the paper in the next few days. We will update Figure 3 to display the top 10 outliers in terms of the absolute value of the difference between the best-fit curve and the residual.
>
> **Q6: Starling and other models in Fig 5.**
> Yes, thank you for pointing this out. This is because some of the figures were created at the time of the original LiveBench release, and other figures were created at the time of the most recent LiveBench release. We are currently updating all figures and tables in the paper to be consistent, and we will finish in the next few days.
>
> **Q7: Citing all models**
> We had referred to Figure 5 to save space, mistakenly neglecting to cite the remaining models. We have now added all citations in a new table, Table 4.
>
> **Q8: The answer format**
> Thank you for pointing this out. We have now added a discussion of our grading methodology to Appendix A.4. In summary: we are very liberal with accepting different answer formats and handling edge cases in our grading function (using the rule of thumb that if a human reading the LLM's answer would be able to comprehend the answer (and assuming it is correct), then the answer should be marked correct. We regularly check the LLM output when running new LLMs on LiveBench, in order to ensure fairness and accuracy in the scoring (since, occasionally a new LLM will introduce a new answer-formatting edge case).
>
> We agree with you that another answer format may be better, e.g., it could cause fewer edge cases. For the new update to the LiveBench questions, we used XML tags. See our full discussion in Appendix A.4.
>
> **Q9: LLM Judge rubric.**
> Yes, this is a typo that we have now corrected. The judge prompt is based off of the original FastChat judge prompt, which is on a 10-point scale. We later changed the prompt to have a binary scale in order to run our LLM judging experiments. The sentence changed to the following:
> After providing your explanation, you must rate the response as either 1 (correct) or 0 (incorrect) by strictly following this format: \"[[rating]]\", for example: \"Rating: [[1]]\"
>
> Thank you very much, once again, for your excellent comments. We respectfully ask that if we have addressed your points or improved your view of our work, to please consider updating your score. If not, please let us know what can be further improved; we are happy to continue the discussion for the final couple days of the discussion period.

---

> > ### Comment · Area_Chair_Gerd · 2024-11-26
> >
> > Dear reviewer uo5E,
> >
> > Could you please response to authors' rebuttal and see if you would like to update your review? Thanks very much!
> >
> > AC

---

> > ### Comment · Reviewer_uo5E · 2024-11-28
> > **Official Comment by Reviewer**
> >
> > I appreciate the authors addressing my concerns.
> >
> > Overall, I believe this benchmark has potential to help the research community. I personally agree with the authors that "impact and value to the community" is more important than "technical novelty" in this datasets and benchmarks category. I believe there is value in the author's efforts to mitigate contamination for the benchmark.
> >
> > I would like to note that "for many types of hard math and reasoning questions, if an LLM struggles to answer the question, then it will also struggle to determine whether or not a given answer to that question is correct." is a claim that should be made with caution. While the authors acknowledge there are exceptions, I suspect these exceptions may actually be the norm--- especially when checking for incorrectness.
> >
> > While the revised section 3.4 is far improved, the authors still have considerable focus in their paper arguing against the efficacy of LLM judges. I don't think the collected numbers of 10% - 46% error rate are particularly meaningful: the performance of LLM judges are so dependent on the surrounding system (prompts, reference answers, pairwise etc) to the point where this experiment feels a little strawman-ish. That being said, I recognize that an in-depth experiment on this area simply is not the focus of this paper. It is my honest opinion that measuring the quality of LLM judgments is orthogonal to the goals of the paper. My understanding is that LiveBench doesn't need LLM-as-a-judge because the prompts are closed-ended and verifiable--- it is simply unnecessary use an LLM judge and fundamentally incompatible with the philosophy of the benchmark... why would anyone use an LLM judge if they have access to verifiable ground truth answers? Therefore, I feel that the authors experiments are neither rigorous enough to make any generalized statements about LLM judges nor particularly relevant to LiveBench specifically. Given this, what really is the purpose of this experiment?
> >
> > Ultimately, the content and focus of the paper is the authors' decision. However, I hope the authors genuinely consider the points above.
> >
> > Some very minor things I noticed while reading through the revision:
> > 1. Some goofy text artifacts on lines 971 and 1012 (dots on top of some characters)
> > 2. Line 1788/1789 it says "5Yes" instead of "Yes"
> > 3. Between lines 1795 and 1796 there is a missing "-" after "Yes"
> >
> > Despite my concerns with section 3.4, I believe the underlying benchmark is useful and well-constructed. The revisions made help make the paper more cohesive overall. The claims on "contamination-free" have been adequately addressed.
> >
> > I will raise my score. **This under the assumption that the authors with faithfully follow through on their planned revisions for W4, Q5, and Q6. Please make sure these changes happen.**

---

> > > ### Author Response · Authors · 2024-11-28
> > > **Second reply to uo5E, Part 1**
> > >
> > > Thank you very much for your response. We are glad that you find that our work has potential to help the research community. We agree with your thoughts on the LLM judge ablation, and we have progress on the other remaining tasks above. Here are more details:
> > >
> > > **W4.**
> > > We have now finished the comparisons of LiveBench to Chatbot Arena, Chatbot Arena style-controlled, Arena Hard, and Arena Hard style-controlled. We have not yet created the new bar plots to update Figure 4 in the paper, but once we do that, we will also replace the sentence in the paper that says “the difference [between LiveBench and Arena Hard] is possibly due to style”, now instead pointing to the correlation matrix that has results for both normal and style-controlled. Here are the new results (raw results, and the correlation matrix):
> > >
> > > | Model                                   | LiveBench | Chatbot Arena | Chatbot Arena (Style Controlled) | Arena Hard | Arena Hard (Style Controlled) |
> > > |-----------------------------------------|-----------|---------------|-----------------------------------|------------|-------------------------------|
> > > | o1-preview-2024-09-12                   | 64.74     | 1334          | 1301                              | 90.4       | 81.7                         |
> > > | claude-3-5-sonnet-20241022              | 58.49     | 1282          | 1283                              | 85.2       | 86.4                         |
> > > | claude-3-5-sonnet-20240620              | 58.22     | 1268          | 1259                              | 79.3       | 82.2                         |
> > > | o1-mini-2024-09-12                      | 56.66     | 1308          | 1258                              | 92         | 79.3                         |
> > > | gpt-4o-2024-08-06                       | 53.77     | 1265          | 1249                              | 77.9       | 71.1                         |
> > > | gemini-1.5-pro-002                      | 53.41     | 1301          | 1266                              | 72         | 62.7                         |
> > > | gpt-4o-2024-05-13                       | 52.6      | 1285          | 1262                              | 79.2       | 69.9                         |
> > > | meta-llama-3.1-405b-instruct-turbo      | 51.12     | 1266          | 1251                              | 69.3       | 67.1                         |
> > > | gpt-4-turbo-2024-04-09                  | 49.6      | 1256          | 1241                              | 82.6       | 74.3                         |
> > > | claude-3-opus-20240229                  | 48.06     | 1248          | 1238                              | 60.4       | 65.5                         |
> > > | gemini-1.5-flash-002                    | 47.62     | 1271          | 1228                              | 49.6       | 39.9                         |
> > > | mistral-large-2407                      | 47.03     | 1251          | 1230                              | 70.4       | 63.1                         |
> > > | gpt-4-0125-preview                      | 44.8      | 1245          | 1230                              | 78         | 73.6                         |
> > > | meta-llama-3.1-70b-instruct-turbo       | 43.21     | 1247          | 1213                              | 55.7       | 51.8                         |
> > > | gpt-4o-mini-2024-07-18                  | 40.25     | 1273          | 1233                              | 74.9       | 64                           |
> > > | claude-3-haiku-20240307                 | 33.22     | 1179          | 1177                              | 41.5       | 45.4                         |
> > > | meta-llama-3.1-8b-instruct-turbo        | 25.03     | 1175          | 1131                              | 21.3       | 18.3                         |
> > >
> > >
> > > | Correlation Matrix                     | LiveBench | ChatBot Arena | Chatbot Arena (Style Controlled) | Arena Hard | Arena Hard (Style Controlled) |
> > > |----------------------------------------|-----------|---------------|-----------------------------------|------------|-------------------------------|
> > > | **LiveBench**                          | 1.000     | 0.890         | 0.963                             | 0.857      | 0.852                         |
> > > | **ChatBot Arena**                      | 0.890     | 1.000         | 0.923                             | 0.826      | 0.714                         |
> > > | **Chatbot Arena (Style Controlled)**   | 0.963     | 0.923         | 1.000                             | 0.896      | 0.878                         |
> > > | **Arena Hard**                         | 0.857     | 0.826         | 0.896                             | 1.000      | 0.955                         |
> > > | **Arena Hard (Style Controlled)**      | 0.852     | 0.714         | 0.878                             | 0.955      | 1.000                         |

---

> > > ### Author Response · Authors · 2024-11-28
> > > **Second reply to uo5E, Part 2**
> > >
> > > **Q5 and Q6.**
> > > We updated Figure 1 and Table 1 with the newest question release from November, and the most recent models (which are also listed in Table 4). We are still working on updating Figures 2-4 now, as well as many of the figures and tables in the appendix. We will continue to let you know our progress in updating all of the figures and tables to be consistent.
> > >
> > > **LLM Judging ablations.**
> > > We agree with your comments that a different experimental setup, e.g., a more detailed or more tailored prompt, could affect the results of our LLM judge ablation, and we also agree that this is orthogonal to our paper.
> > >
> > > We have now removed the mention of this experiment from the intro. (The deadline to update the pdf is now, but with more time, we will also modify the abstract and intro to focus less on “LLM judging has pitfalls, therefore we introduce LiveBench” and more on “we introduce LiveBench which has ground-truth answers [implying that LLM judging is unnecessary].”)
> > >
> > > We moved most of the discussion on the LLM judge ablation to the appendix, and we described it as a preliminary study. In the appendix, we change the language so as not to claim that our experiment is definitive, but that it is a preliminary study and that there might be another experimental setup (e.g., more tailored prompt) that causes the LLM judges to do better.
> > >
> > > **Minor edits**
> > > Thank you for pointing these out. We have now corrected them all.
> > >
> > > Thank you once again, and please let us know if you have more follow-up questions or suggestions.

---

> > > > ### Comment · Reviewer_uo5E · 2024-11-30
> > > > **Reviewer Reply to Authors**
> > > >
> > > > Thank you for the updates. The new tables in W4 are very informative, I believe they better contextualize the evaluation signal LiveBench captures.
> > > > I appreciate the authors heeding my concerns on the LLM judge ablations.
> > > >
> > > > Please keep me updated as the last few figures get updated. Thanks!

---

> > > > > ### Author Response · Authors · 2024-12-01
> > > > > **Third reply to uo5E, Part 1**
> > > > >
> > > > > We are glad that you appreciate the updates regarding benchmark comparison and LLM judge ablations. Here is an update on the remaining figures and tables in the paper. Note that while we created the new figures/tables, we cannot update the pdf, so we will describe the updates here.
> > > > >
> > > > > **Fig 1, Table 1** - Already updated in the current pdf.
> > > > >
> > > > > **Fig 2** Correlations among categories and groups in LiveBench For each pair of categories in LiveBench, we compute the Pearson correlation coefficient based on the results for all models.
> > > > > | | math | coding | language | reasoning | data_analysis | instruction_following |
> > > > > |---------------------|-------|---------|-----------|------------|----------------|---------------------|
> > > > > | math | 1.000 | 0.869 | 0.777 | 0.885 | 0.852 | 0.746 |
> > > > > | coding | 0.869 | 1.000 | 0.736 | 0.808 | 0.778 | 0.651 |
> > > > > | language | 0.777 | 0.736 | 1.000 | 0.811 | 0.888 | 0.674 |
> > > > > | reasoning | 0.885 | 0.808 | 0.811 | 1.000 | 0.860 | 0.735 |
> > > > > | data_analysis | 0.852 | 0.778 | 0.888 | 0.860 | 1.000 | 0.714 |
> > > > > | instruction_following | 0.746 | 0.651 | 0.674 | 0.735 | 0.714 | 1.000 |
> > > > >
> > > > > Note: Fig 2 (right) is too large to display in table form here. The noticeable difference between the previous Fig 2 (right) and the updated version is the order of the tasks: web_of_lies_v2, math_comp, tablejoin, zebra_puzzle, coding_completion, cta, plot_unscrambling, LCB_generation, connections, AMPS_Hard, olympiad, summarize, paraphrase, simplify, typos, story_generation, spatial, tablereformat.
> > > > >
> > > > > Noticeably, due to the update described in our global comment, zebra_puzzle went from the task that is lowest correlated with the average LiveBench score, to the 3rd most correlated with the average LiveBench score, after creating a wider variety of difficulties and giving partial credit. See also our updated Table 6 below. Correlation with average LiveBench score may have different contributing factors, such as the amount of noise or the uniqueness of the task. We believe that in this case, the increase for zebra puzzle points to decreased noise, since the changes to zebra puzzle were to give a much wider distribution of difficulty, and to change the prompt such that each question allows for partial credit.
> > > > >
> > > > > **Fig 3** Reasoning performance vs. average performance. Since we cannot show the plot, we give a table of the top ten outliers, computed as the largest absolute difference between the model’s score and the value of the best fit line.
> > > > > | model | residual | sign |
> > > > > |---------|-----------|------|
> > > > > | o1-mini-2024-09-12 | 16.2680 | + |
> > > > > | phi-3-mini-4k-instruct | 10.6320 | + |
> > > > > | claude-3-5-haiku-20241022 | -10.0715 | - |
> > > > > | gemini-exp-1121 | -9.3577 | - |
> > > > > | gemma-2-27b-it | -8.5776 | - |
> > > > > | gpt-4o-2024-11-20 | 7.2868 | + |
> > > > > | phi-3-mini-128k-instruct | 7.0328 | + |
> > > > > | gemma-2-9b-it | -6.0608 | - |
> > > > > | gemini-1.5-flash-001 | -6.0274 | - |
> > > > > | claude-3-opus-20240229 | -5.8654 | - |
> > > > >
> > > > > Here is the same table for instruction following (Fig 3, right).
> > > > > | model | residual | sign |
> > > > > |---------|-----------|------|
> > > > > | gemini-1.5-flash-002 | 12.8877 | + |
> > > > > | gemini-1.5-flash-8b-exp-0827 | 12.2437 | + |
> > > > > | phi-3-mini-4k-instruct | -10.7181 | - |
> > > > > | gemini-1.5-flash-exp-0827 | 9.5700 | + |
> > > > > | step-2-16k-202411 | 8.6686 | + |
> > > > > | phi-3-mini-128k-instruct | -8.5855 | - |
> > > > > | command-r-plus-04-2024 | 8.4812 | + |
> > > > > | gemini-exp-1121 | 8.2303 | + |
> > > > > | o1-mini-2024-09-12 | -6.9661 | - |
> > > > > | meta-llama-3.1-70b-instruct-turbo | 6.9083 | + |
> > > > >
> > > > > **Fig 5, Table 2 (extensions of Fig 1, Table 1, respectively)** - In global comment
> > > > >
> > > > > **Table 3, 8, 9, 12** - These do not need to be updated.
> > > > >
> > > > > **Figs 4 & 6; Tables 4, 13, 14** - updated during this rebuttal period, so they are already updated.
> > > > >
> > > > > **Table 5**: Pearson correlation coefficient and std. error for each category compared to the overall average LiveBench score, computed using data from all models.
> > > > > | Category | Correlation | Std Error |
> > > > > |----------|-------------|-----------|
> > > > > | math | 0.9477 | 0.0643 |
> > > > > | reasoning | 0.9439 | 0.0709 |
> > > > > | data_analysis | 0.9315 | 0.0489 |
> > > > > | coding | 0.8970 | 0.0840 |
> > > > > | language | 0.8970 | 0.0831 |
> > > > > | instruction_following | 0.8174 | 0.0811 |
> > > > >
> > > > > **Table 6**: Pearson correlation coefficient and std. error for each task compared to the overall average LiveBench score, computed using data from all models.
> > > > > | Category | Correlation | Std Error |
> > > > > |----------|-------------|-----------|
> > > > > | web_of_lies_v2 | 0.9136 | 0.1454 |
> > > > > | math_comp | 0.9035 | 0.0986 |
> > > > > | tablejoin | 0.8984 | 0.0961 |
> > > > > | zebra_puzzle | 0.8853 | 0.0935 |
> > > > > | coding_completion | 0.8623 | 0.1212 |
> > > > > | cta | 0.8556 | 0.0486 |
> > > > > | plot_unscrambling | 0.8553 | 0.0881 |
> > > > > | LCB_generation | 0.8550 | 0.0816 |
> > > > > | connections | 0.8368 | 0.1188 |
> > > > > | AMPS_Hard | 0.8245 | 0.1357 |
> > > > > | olympiad | 0.8192 | 0.1171 |
> > > > > | summarize | 0.8044 | 0.0904 |
> > > > > | paraphrase | 0.7884 | 0.0953 |
> > > > > | simplify | 0.7765 | 0.0829 |
> > > > > | typos | 0.7722 | 0.1474 |
> > > > > | story_generation | 0.7443 | 0.1018 |
> > > > > | spatial | 0.7294 | 0.0968 |
> > > > > | tablereformat | 0.6840 | 0.1058 |

---

> > > > > ### Author Response · Authors · 2024-12-01
> > > > > **Third reply to uo5E, Part 2**
> > > > >
> > > > > **Table 7**: Relative best and worst task for each model, computed as the tasks with the highest and lowest residuals of the best fit line vs. overall LiveBench performance, for each model.
> > > > > | Model | Best | Worst |
> > > > > |---------|---------|---------|
> > > > > | o1-preview-2024-09-12 | connections | coding_completion |
> > > > > | claude-3-5-sonnet-20241022 | coding_completion | web_of_lies_v2 |
> > > > > | claude-3-5-sonnet-20240620 | olympiad | math_comp |
> > > > > | o1-mini-2024-09-12 | zebra_puzzle | coding_completion |
> > > > > | gemini-exp-1114 | AMPS_Hard | typos |
> > > > > | gemini-exp-1121 | story_generation | web_of_lies_v2 |
> > > > > | step-2-16k-202411 | paraphrase | LCB_generation |
> > > > > | gpt-4o-2024-08-06 | spatial | web_of_lies_v2 |
> > > > > | gemini-1.5-pro-002 | olympiad | web_of_lies_v2 |
> > > > > | gpt-4o-2024-05-13 | typos | tablejoin |
> > > > > | gemini-1.5-pro-exp-0827 | olympiad | LCB_generation |
> > > > > | meta-llama-3.1-405b-instruct-turbo | web_of_lies_v2 | connections |
> > > > > | gpt-4o-2024-11-20 | typos | olympiad |
> > > > > | dracarys2-72b-instruct | coding_completion | typos |
> > > > > | chatgpt-4o-latest-0903 | spatial | olympiad |
> > > > > | gpt-4-turbo-2024-04-09 | connections | simplify |
> > > > > | claude-3-opus-20240229 | typos | web_of_lies_v2 |
> > > > > | gemini-1.5-flash-002 | summarize | typos |
> > > > > | mistral-large-2407 | olympiad | tablejoin |
> > > > > | qwen2.5-coder-32b-instruct | LCB_generation | typos |
> > > > > | gpt-4-0125-preview | tablereformat | olympiad |
> > > > > | gemini-1.5-flash-exp-0827 | web_of_lies_v2 | AMPS_Hard |
> > > > > | gemini-1.5-pro-001 | typos | web_of_lies_v2 |
> > > > > | meta-llama-3.1-70b-instruct-turbo | tablereformat | zebra_puzzle |
> > > > > | claude-3-5-haiku-20241022 | coding_completion | web_of_lies_v2 |
> > > > > | gpt-4o-mini-2024-07-18 | AMPS_Hard | olympiad |
> > > > > | gemini-1.5-flash-001 | typos | web_of_lies_v2 |
> > > > > | gemma-2-27b-it | tablereformat | web_of_lies_v2 |
> > > > > | gemini-1.5-flash-8b-exp-0827 | paraphrase | plot_unscrambling |
> > > > > | qwen2.5-7b-instruct-turbo | AMPS_Hard | typos |
> > > > > | claude-3-haiku-20240307 | typos | web_of_lies_v2 |
> > > > > | mixtral-8x22b-instruct-v0.1 | coding_completion | tablereformat |
> > > > > | command-r-plus-08-2024 | typos | coding_completion |
> > > > > | gemma-2-9b-it | typos | spatial |
> > > > > | mistral-small-2402 | web_of_lies_v2 | zebra_puzzle |
> > > > > | command-r-08-2024 | paraphrase | AMPS_Hard |
> > > > > | command-r-plus-04-2024 | simplify | tablereformat |
> > > > > | meta-llama-3.1-8b-instruct-turbo | typos | spatial |
> > > > > | phi-3-mini-128k-instruct | spatial | story_generation |
> > > > > | phi-3-mini-4k-instruct | spatial | simplify |
> > > > >
> > > > > **Tables 10, 11** - the updated olympiad task now uses hard difficulty for all problems, so these analyses no longer apply. We will keep these tables with a note that it’s from the original olympiad task.
> > > > >
> > > > > **Table 15, 16** - We will update these later this month.
> > > > >
> > > > > Thank you once again for all of your suggestions durring the response period!

---

> > > > > > ### Comment · Reviewer_uo5E · 2024-12-02
> > > > > > **Official Comment by Reviewer**
> > > > > >
> > > > > > Thank you for the updates. It appears the authors have followed through on their planned revisions for W4, Q5, and Q6. My concerns have been addressed; I will maintain my raised score.

---

### Author Response · Authors · 2024-11-25

We thank all of the reviewers for their valuable feedback. Our work introduces LiveBench, a challenging benchmark that is regularly updated in order to limit contamination.  We appreciate that the reviewers found our work to be unique and a great contribution to LLM research (uo5E), with diverse, challenging tasks (uo5E, G9nN, L2M4). We also appreciate that reviewers found our work to be clearly written (uo5E, L2M4), with high reproducibility and transparency (G9nN).

We believe that LiveBench fills an important role in the research community: it is a consistent measure of hard reasoning questions, similar to MMLU and GSM8K, or BigBench-Hard, while being minimally contaminated. There are other benchmarks with limited or no contamination, but they have substantial differences such as lack of transparency, being expensive to update, or using LLM judging. LiveBench occupies an important position of being a contamination-limited, reasoning-focused leaderboard, and new models can be added very quickly to the leaderboard.

We highlight all of the new additions based on the reviewers’ suggestions
- We added new sections to the paper
  - We added Section A.4: Grading Methodology.
  - We added Section A.5: Quality Control for New Models.
  - We added an expanded section on the question update policy: A.6.
 - We updated the language in the title and throughout the paper from claiming LiveBench is “contamination-free” to “contamination-limited” (note: we still argue that LiveBench has high impact). We also add a new discussion on the different types of contamination to Section A.7.
- We updated and added new figures and tables
  - Table 4 - Citations for all models in LiveBench.
  - Table 13 - Describes the data source for each LiveBench task.
  - Table 14 - Describes the time to update each LiveBench task.
- We updated and clarified additional parts of the paper based on reviewer feedback.

We also have preliminary results for the next update to LiveBench. We updated all four instruction following tasks, the connections task, and the Zebra puzzles task. For the instruction following tasks and the connections task, we created all new questions using new information sources (new news articles for instruction following, and new daily connections puzzles). We also made the distribution of problems harder: adding more instructions per question to the instruction following tasks, and adding more 4 by 4 connections puzzles. For Zebra puzzles, we significantly increased the difficulty, due to o1-preview and o1-mini achieving high scores on the old version. The new version has a wider range of both puzzle size and clue-difficulty, and asks three questions about the solution rather than one, making it harder to guess. In addition, we give partial credit now, making the task have less noise. The following are preliminary results, with the full results to come very soon (before the end of the rebuttal period).

| Model | Connections | Simplify | Story Generation | Summarize | Zebra Puzzle |
|---|--:|--:|--:|--:|--:|
| claude-3-5-haiku-20241022 | 26.17 | 70.65 | 59.00 | 58.65 | 22.25 |
| claude-3-5-sonnet-20241022 | 43.17 | 72.67 | 68.67 | 69.05 | 44.00 |
| gpt-4o-2024-11-20 | 36.67 | 67.75 | 66.25 | 63.10 | 47.00 |
| gpt-4o-mini-2024-07-18 | 16.50 | 60.73 | 56.13 | 52.98 | 28.25 |
| o1-mini-2024-09-12 | 59.33 | 73.00 | 63.00 | 59.00 | 67.00 |
| o1-preview-2024-09-12 | 86.67 | 78.08 | 71.58 | 69.63 | 66.25 |
| average | 44.75 | 70.48 | 64.11 | 62.07 | 45.79 |

Thank you once again, and we are happy to answer any final questions!

---

> ### Author Response · Authors · 2024-12-01
>
> Thank you for replying and continue to engage in discussion regarding our paper. We would like to summarize the recent updates we made to the paper.
>
> - LiveBench update; details below.
> - New comparisons with style-controlled Chatbot Arena and Arena Hard: see [here](https://openreview.net/forum?id=sKYHBTAxVa&noteId=NwDAgNocps).
> - All figures and tables in the paper are updated based on the latest LiveBench data: see [here](https://openreview.net/forum?id=sKYHBTAxVa&noteId=yuTuP40ccy).
> - Changed the wording regarding the LLM judge ablation: see [here](https://openreview.net/forum?id=sKYHBTAxVa&noteId=3egUK1dOWq).
>
> **LiveBench update.**
> We have updated both the models in LiveBench, as well as the questions. Here is the list of new models that we have added since Oct 1:
> `claude-3-5-haiku-20241022, claude-3-5-sonnet-20241022, dracarys2-72b-instruct, gemini-exp-1114, gemini-exp-1121, gpt-4o-2024-11-20, grok-2, grok-2-mini, llama-3.1-nemotron-70b-instruct, qwen-2.5-7b-instruct-turbo, qwen2.5-Coder-32B-Instruct, step-2-16k-202411`.
>
> We have also updated the LiveBench questions. We updated the connections task, the zebra puzzles task, and all four instruction following tasks (more details below). This consists of 300 updated questions total. Here is the updated leaderboard (top 15 models):
>
> | model | average | coding | data_analysis | instruction_following | language | math | reasoning |
> |---------|----------|---------|----------------|----------------------|-----------|------|-----------|
> | o1-preview-2024-09-12 | 64.7 | 50.8 | **64.0** | 74.6 | **68.7** | **62.9** | 67.4 |
> | claude-3-5-sonnet-20241022 | 58.5 | **67.1** | 52.8 | 69.3 | 53.8 | 51.3 | 56.7 |
> | claude-3-5-sonnet-20240620 | 58.2 | 60.8 | 56.7 | 68.0 | 53.2 | 53.3 | 57.2 |
> | o1-mini-2024-09-12 | 56.7 | 48.1 | 54.1 | 65.4 | 40.9 | 59.2 | **72.3** |
> | gemini-exp-1114 | 56.0 | 52.4 | 57.5 | 77.1 | 38.7 | 54.9 | 55.7 |
> | gemini-exp-1121 | 56.0 | 50.4 | 57.0 | **80.1** | 40.0 | 62.8 | 45.8 |
> | step-2-16k-202411 | 55.1 | 46.9 | 54.9 | 79.9 | 44.5 | 48.9 | 55.5 |
> | gpt-4o-2024-08-06 | 53.8 | 51.4 | 52.9 | 68.6 | 47.6 | 48.2 | 53.9 |
> | gemini-1.5-pro-002 | 53.4 | 48.8 | 52.3 | 70.8 | 43.3 | 57.4 | 47.9 |
> | gpt-4o-2024-05-13 | 52.6 | 49.4 | 52.4 | 68.2 | 50.0 | 46.0 | 49.6 |
> | gemini-1.5-pro-exp-0827 | 52.4 | 40.9 | 50.8 | 69.3 | 46.1 | 56.1 | 50.9 |
> | meta-llama-3.1-405b-instruct-turbo | 51.1 | 43.8 | 53.5 | 72.8 | 43.2 | 40.5 | 52.8 |
> | gpt-4o-2024-11-20 | 50.6 | 46.1 | 47.2 | 64.9 | 47.4 | 42.5 | 55.7 |
> | dracarys2-72b-instruct | 50.1 | 56.6 | 49.1 | 65.2 | 33.5 | 50.6 | 45.8 |
> | chatgpt-4o-latest-0903 | 50.1 | 47.4 | 48.7 | 66.4 | 45.3 | 42.1 | 50.5 |
>
> Here are more details for each of the tasks.
>
> Zebra puzzles: We replaced all 50 zebra puzzle questions. We significantly changed the question generation script to give a larger variety of puzzle lengths and puzzle difficulties. The mean decreased by 7.68 percentage points. The biggest performance drops were the following models:
> | Model                                  | Difference  |
> |----------------------------------------|-------------|
> | gemma-2-27b-it                         | -29.75      |
> | command-r-plus-04-2024                 | -26.50      |
> | phi-3-mini-128k-instruct               | -25.25      |
> | mistral-small-2402                     | -24.25      |
> | command-r-plus-08-2024                 | -17.75      |
>
> Connections: We replaced all 50 connections questions. We replaced the questions with new questions based on connections puzzles from the range Oct 4 to Nov 24, and we updated the distribution of puzzle sizes to make the task harder. The mean decreased by 8.80 percentage points. The biggest performance drops were the following models:
> | Model                                  | Difference  |
> |----------------------------------------|-------------|
> | meta-llama-3.1-70b-instruct-turbo      | -24.167     |
> | gpt-4o-2024-08-06                      | -20.333     |
> | gpt-4o-mini-2024-07-18                 | -20.000     |
> | meta-llama-3.1-405b-instruct-turbo     | -19.834     |
> | gemini-exp-1114                        | -18.667     |
>
> Instruction Following: we replaced all four tasks: 200 questions total. We replaced the source data (news articles) to news articles from the range Nov 1 to Nov 24. We also made the task harder by increasing the average number of instructions per question. The mean decreased by 7.39  percentage points. The biggest performance drops were the following models:
> | Model                                  | Difference  |
> |----------------------------------------|-------------|
> | phi-3-mini-4k-instruct                 | -14.9       |
> | phi-3-mini-128k-instruct               | -12.3       |
> | command-r-plus-04-2024                 | -12.0       |
> | mixtral-8x22b-instruct-v0.1            | -10.9       |
> | gpt-4-turbo-2024-04-09                 | -10.6       |
>
> Please let us know if you have any final thoughts or suggestions. Thank you!

---

### Meta-Review · Area_Chair_Gerd · 2024-12-23

**Metareview:**

All reviewers agreed that this is a great paper and should be included in the conference: this paper provides a nice way of  minimizing the test set contamination with frequently-updated questions from recent information sources and procedural question generation techniques.

Strength:
1. Well-rounded verifiable benchmarks that cover a large and diverse set of of tasks.
2. Paper is clearly written.

Weakness:
Regex matching for validation might be limited.

**Additional Comments On Reviewer Discussion:**

All reviewers actively participated the discussion during the rebuttal and the pointes raised by the reviewers were adequately addressed.

---

### Decision · Program_Chairs · 2025-01-22

Accept (Spotlight)